# Heeding the Inner Voice: Aligning Control-Net Training via Intermediate Features Feedback

## Abstract

Despite significant progress in text-to-image diffusion models, achieving precise spatial control over generated outputs remains challenging. One of the popular approaches for this task is ControlNet, which introduces an auxiliary conditioning module into the architecture. To improve alignment of the generation image and control, ControlNet++ proposes a cycle consistency loss to refine correspondence between controls and outputs, but restricts its application to the final denoising steps, while the main structure is introduced at an early stage of generation. To address this issue, we suggest **InnerControl** – a training strategy that enforces spatial consistency across all diffusion steps. Specifically, we train lightweight control prediction probes — small convolutional networks — to reconstruct input control signals (e.g., edges, depth) from intermediate UNet features at every denoising step. We prove the efficiency of such models to extract signals even from very noisy latents and utilize these models to generate pseudo ground truth controls during training. Suggested approach enables alignment loss that minimizes the difference between predicted and target condition throughout the whole diffusion process. Our experiments demonstrate that our method improves control alignment and fidelity of generation. By integrating this loss with established training techniques (e.g., ControlNet++), we achieve high performance across different condition methods such as edge, segmentation and depth conditions.

## 1 Introduction

Recent advances in diffusion models (Ho et al., 2020; Song et al., 2020; Sohl-Dickstein et al., 2015; Dhariwal & Nichol, 2021) have significantly improved the quality and diversity of text-to-image (T2I) generation, enabling models to produce images that closely match input textual prompts Nichol et al. (2021); Ramesh et al. (2021); Rombach et al. (2022); Saharia et al. (2022). Despite this progress, achieving **precise spatial control** over generated images remains a key challenge Hu et al. (2023); Ye et al. (2023); Huang et al. (2023).

To address this issue, methods such as ControlNet Zhang et al. (2023) and T2I-Adapter Mou et al. (2024) introduce conditional mechanisms to guide the generation process using control signals (e.g., edge maps, depth maps, segmentation masks). Follow-up work has aimed to improve ControlNet performance through architectural enhancements Zavadski et al. (2024), unified conditioning Qin et al. (2023); Zhao et al. (2023), and efficient adaptation to new conditions Xu et al. (2024). However, these approaches often suffer from **inconsistencies between the input control signals and the final generated output.**

Recent methods aim to improve control fidelity by introducing additional supervision mechanisms during training. For example, ControlNet++ Li et al. (2024) reduces the discrepancy between the input control signal and the generated image by applying reward losses that penalize inconsistencies between the generated output and the extracted control signals (e.g., edges or depth). In contrast, CTRL-U Zhang et al. (2024) proposes an alternative approach based on uncertainty-aware reward modeling, which aims to mitigate the negative impact of inaccurate or noisy feedback from reward models. While both methods demonstrate improved control alignment, they primarily operate at

**late denoising steps**, despite evidence that **spatial structure emerges early in the diffusion process** Chen et al.; Baranchuk et al. (2021).

However, extending the reward losses to earlier steps leads to a significant decrease in the generated image quality, producing visible artifacts on the generated images. These poor results are probably caused by inefficient signal extraction in the early sampling steps, producing inaccurate signals for loss calculations. This analysis highlights a critical limitation of the suggested approaches, making them applicable only to the late generation steps.

To address temporal misalignment in prior methods, we introduce **InnerControl** – a novel training strategy that enforces consistency between input control (e.g., edges, depth, segmentation) and signals extracted from **intermediate diffusion features across the entire denoising trajectory**. Our approach is motivated by recent findings that demonstrate the utility of diffusion features for vision tasks such as depth estimation, semantic segmentation, and classification Baranchuk et al. (2021); Hedlin et al. (2023); Namekata et al. (2024). Building on this, we propose to use lightweight convolutional networks to extract control signals directly from UNet decoder features.

Drawing inspiration from Readout Guidance Luo et al. (2024), which employs timestep-conditioned architectures for discriminative tasks and demonstrates the effectiveness of these models during the early stages of denoising—when spatial structure primarily emerges Chen et al.—we utilize these estimation models to introduce an additional penalty during ControlNet training. This penalty explicitly enforces spatial alignment throughout the entire generation process. Our findings indicate that **InnerControl** enhances the previous reward training approach, improving control alignment while preserving perceptual quality.

Our core contributions are:

- **Early-stage control alignment:** we propose a novel training objective that enforces consistency between the input control signal (e.g., edge, depth and segmentation maps) and signals extracted from intermediate diffusion features across the entire denoising process, including early stages where structural content begins to emerge.

- **Enhanced controllability:** our training strategy improves upon existing reward-based approaches, achieving stronger control alignment and higher image quality across diverse spatial control tasks, such as depth, edge and segmentation guidance.

## 2 RELATED WORK

### 2.1 CONTROLLABLE TEXT-TO-IMAGE DIFFUSION MODELS

Diffusion models Ho et al. (2020); Song et al. (2020); Sohl-Dickstein et al. (2015); Dhariwal & Nichol (2021) have achieved remarkable success in generating high-quality, diverse images conditioned on text prompts Nichol et al. (2021); Ramesh et al. (2021); Rombach et al. (2022); Saharia et al. (2022); Balaji et al. (2022); Ramesh et al. (2022). However, traditional approaches rely solely on textual guidance, limiting precise spatial control over generated outputs. To address this, several methods introduce spatial control signals without retraining the entire diffusion pipeline. ControlNet Zhang et al. (2023) augments pretrained diffusion models with a duplicate encoder and zero-convolution layers, enabling stable training and alignment with diverse spatial conditions (e.g., edges, depth, segmentation masks). Similarly, T2I-Adapter Mou et al. (2024) employs lightweight adapter modules to bridge internal text-to-image representations with external control inputs. Subsequent works refine these designs for greater efficiency Zavadski et al. (2024); Cao et al. (2025), extend them into unified frameworks supporting multiple control types Zhao et al. (2023); Qin et al. (2023), or improve adaptability to novel control signals Xu et al. (2024) and advanced backbones Lin et al. (2024); Ran et al. (2024). Despite these advances, ensuring consistent alignment between generated outputs and conditioning signals remains challenging.

To address this issue, ControlNet++ Li et al. (2024) introduces an additional reward loss for ControlNet that enhances controllable generation by explicitly optimizing pixel-level cycle consistency between generated images and conditional inputs.Ctrl-U Zhang et al. (2024), on the other hand, introduces uncertainty-aware reward modeling to regularize reward fine-tuning through consistency construction. Specifically, rewards with lower uncertainty are assigned higher loss weights, while

those with higher uncertainty receive reduced weights to accommodate greater variability. However, both approaches primarily target late-stage alignment due to the suggested one-step prediction strategy for signal estimation, thereby neglecting the earlier phases of generation. Meanwhile, prior studies indicate that the main structure emerges during the early stages of generation Chen et al.. This makes consistency throughout the entire generation trajectory essential for preserving fidelity to input conditions. Our approach directly addresses this gap by enforcing alignment at every denoising step.

## 2.2 DIFFUSION MODEL REPRESENTATION

Pretrained text-to-image diffusion models have proven highly effective at extracting semantically rich representations from their internal features, supporting a wide range of discriminative tasks such as segmentation, semantic correspondence, classification, detection, and depth estimation Fundel et al. (2025); Hedlin et al. (2023); Clark & Jaini (2023); Yang & Wang (2023); Xiang et al. (2023). Several studies have analyzed the quality of UNet features across different denoising steps for downstream vision tasks Baranchuk et al. (2021); Chen et al.. Recent efforts extend this line of work by aggregating features across layers and denoising steps to improve discriminative performance, with applications in segmentation Tang et al. (2022); Namekata et al. (2024); Stracke et al. (2024), semantic correspondence Tang et al. (2023); Hedlin et al. (2023); Stracke et al. (2024), classification Li et al. (2023a); Stracke et al. (2024), detection Chen et al. (2023); Stracke et al. (2024), and depth estimation Stracke et al. (2024). However, most existing approaches rely on aggregated features across denoising steps, which may limit their ability to capture task-specific information at each individual stage. To address this limitation, Luo et al. Luo et al. (2024) enhance the Diffusion Hyperfeatures framework Luo et al. (2023) by introducing additional timesteps conditioning.

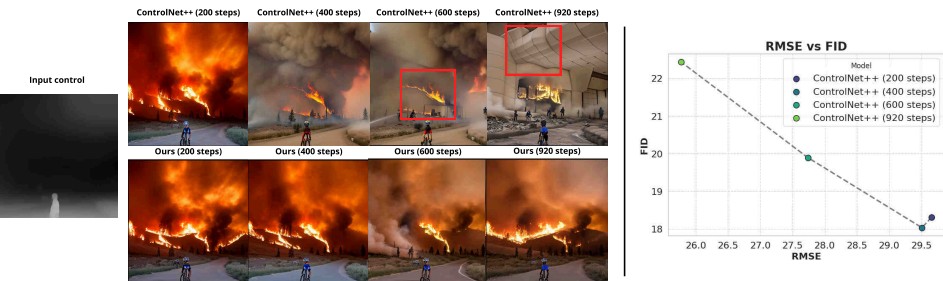

Figure 1: Visualizing the trade-off between control consistency (RMSE) and image fidelity (FID) when extending reward losses to early denoising stages. *Left:* Generated samples illustrate visual artifacts when applying only reward loss (*top*) compared to reward plus alignment loss (*bottom*) during early diffusion steps. *Right:* Quantitative analysis shows the trade-off between control precision (RMSE ↓) and image fidelity (FID ↓), highlighting their inverse relationship.

# 3 PRELIMINARIES

In this section, we introduce the background of diffusion models and spatially controllable generation, followed by an analysis of cycle consistency losses suggested in ControlNet++ Li et al. (2024).

## 3.1 CONTROLLABLE GENERATION

Diffusion models Ho et al. (2020); Song et al. (2020) are a class of generative models that synthesize data by iteratively denoising random noise through a learned reverse process.

The forward process defines a sequence of noise-adding steps that transform the data into isotropic Gaussian noise over $T$ steps.

$$q(x_t|x_{t-1}) = \mathcal{N}\left(x_t; \sqrt{\alpha_t}x_{t-1}, (1-\alpha_t)\mathbf{I}\right) \tag{1}$$

where $\alpha_t \in (0,1)$ is a fixed variance schedule. The reverse process learns to invert this process using a neural network that iteratively denoises samples:

$$p_\theta(x_{t-1}|x_t) = \mathcal{N}\left(x_{t-1}; \mu_\theta(x_t, t), \sigma_t^2\right), \tag{2}$$

where $\mu_\theta(\cdot)$ is a learnable function that approximates the mean of the true posterior.

The standard training objective aims to minimize the noise prediction error:

$$\mathcal{L}_{\text{diff}} = \mathbb{E}_{x_0, \epsilon, t} \left[ \|\epsilon - \epsilon_\theta(x_t, t)\|^2 \right] \tag{3}$$

where $x_t$ is a noisy sample from timestep $t$, $\epsilon \sim \mathcal{N}(0, I)$ and $\epsilon_\theta(x_t, t)$ – learned approximation using another parametrization.

In cases where additional control is required, such as text prompt conditioning $c_{txt}$ and spatial control $c_{spat}$ (e.g., depth maps or edges), the objective can be expressed as:

$$\mathcal{L}_{\text{diff}} = \mathbb{E}_{x_0, \epsilon, t, c_{txt}, c_{spat}} \left[ \|\epsilon - \epsilon_\theta(x_t, t, c_{spat}, c_{txt})\|^2 \right] \tag{4}$$

## 3.2 CONTROLNET++

ControlNet Zhang et al. (2023) is one of the leading methods that utilizes a pretrained text-to-image diffusion model for controllable generation with additional spatial control. While ControlNet is trained using standard diffusion loss defined in Eq. 4, it suffers from inconsistencies between the final predictions and the input controls. To mitigate this issue, ControlNet++ introduces a cycle consistency loss that leverages a discriminative reward model.

Specifically, the method minimizes the discrepancy between the input control $c_{spatial}$ and the corresponding condition $\hat{c}_{spatial}$ extracted from the generated image by the reward model $\mathbb{D}$, where $\hat{c}_{spatial} = \mathbb{D}(x_0)$ and $x_0$ denotes the generated image. Since diffusion models sample timesteps $t \in [999, 0]$ to simulate the denoising process, computing rewards across the full trajectory would require prohibitive gradient accumulation. To address this, ControlNet++ approximates $x_0$ from a noisy sample $x_t$ using a single-step generation process:

$$\begin{aligned} x_0 \approx x_0' = \mathbb{G}(c_{\text{spat}}, c_{\text{txt}}, x_t, t) = \\ = \frac{x_t - \sqrt{1 - \alpha_t}\, \epsilon_\theta(x_t', c_{\text{spat}}, c_{\text{txt}}, t)}{\sqrt{\alpha_t}} \end{aligned} \tag{5}$$

where $\epsilon_\theta(\cdot)$ denotes the network's noise prediction and $G(\cdot)$ represents the single-step denoising operation. The resulting approximation $x_0'$ can then be used for reward fine-tuning:

$$\mathcal{L}_{\text{reward}} = \mathcal{L}\left(c_{spatial}, \mathbb{D}\left[\mathbb{G}(c_{spatial}, c_{txt}, x_t, t)\right]\right) \tag{6}$$

Due to single-step sampling, the authors suggest applying their rewarding loss only on the last 200 steps ($t \in [0, 200]$) of diffusion trajectory sampling.

## 4 METHOD

In this section, we discuss the main limitations of ControlNet++ and introduce our proposed training approach – **InnerControl** – designed to address these issues.

### 4.1 MOTIVATION

As noted earlier, ControlNet++ Li et al. (2024) applies the reward loss only to the final 200 denoising steps ($t \in [0, 200]$) due to its reliance on a single-step prediction strategy. To better understand this limitation, we analyze the trade-off between control consistency and image fidelity when extending the reward loss to earlier denoising stages. Specifically, we train a ControlNet model conditioned on depth maps, applying $\mathcal{L}_{\text{reward}}$ loss over different ranges of denoising steps. Alignment is measured using RMSE, while perceptual quality is evaluated with FID.

Our experiments show that extending the reward loss to earlier steps improves control alignment, as reflected in lower RMSE (Fig. 1). While RMSE improves, the quality of generated images is significantly decreased, leading to an increase in FID (Fig. 1 right). The perceptual metrics degradation can also be observed in the images, where artifacts appear as the reward loss is extended to earlier steps.

As shown in Fig. 1, applying the reward loss up to 400 denoising steps produces good-quality images, but extending it to 600 steps introduces small artifacts, and using it across nearly all steps (e.g., 920) results in severe distortions such as unexpected lines and irregular edges.

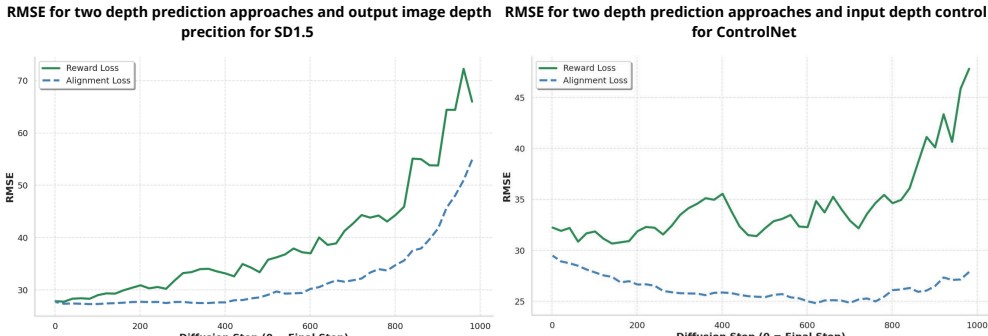

Figure 2: *Left:* RMSE between depth estimated from the final image and DPT depth prediction for single-step predicted images (green) and for depths estimated from intermediate features (blue) for SD1.5 generation. *Right:* RMSE between control depth and DPT depth prediction for single-step predicted images (green) and for depths estimated from intermediate features (blue) for ControlNet.

We hypothesize that this degradation arises from the poor quality of single-step predictions at highly noisy timesteps. In early denoising stages, the single-step image predictions are extremely blurry and fall outside the domain expected by pretrained depth estimators. Fig. 3 illustrates this effect: from around 400 steps onward, single-step predictions become increasingly blurry, leading to unreliable DPT depth estimates. This is further supported in Fig. 2, where RMSE increases substantially at early steps (green line) for both SD1.5 and ControlNet generations. Consequently, the extracted depth maps are inaccurate and misaligned with ground-truth depth map of an image, propagating errors during training.

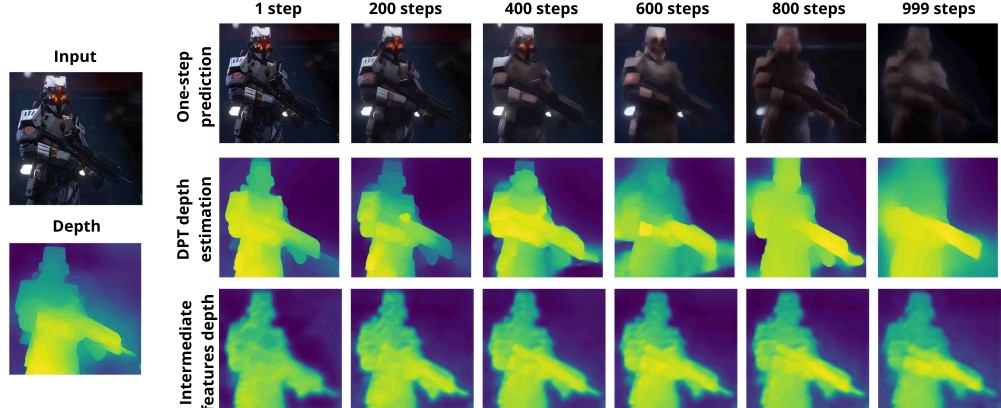

Figure 3: Results of one-step prediction (*top*) at varying noise levels (low → high), with corresponding depth maps predicted by the DPT estimator Ranftl et al. (2021) (*middle*) and by intermediate UNet features (*bottom*).

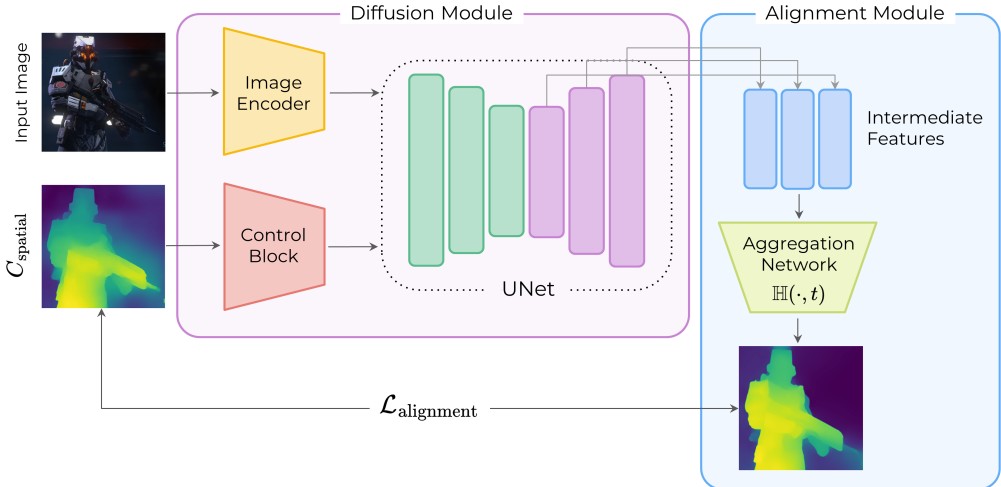

Figure 4: **Pipeline overview.** Schematic illustration of our **InnerControl** framework, emphasizing the integration of the alignment loss. The key difference from ControlNet++ is the **Alignment module**, which processes intermediate features extracted from the UNet decoder. These features are passed through an aggregation network to predict spatial control signals (e.g., depth or edge maps), which are then compared with the input control $c_{spat}$ to enforce consistency at every denoising step.

## 4.2 ALIGNMENT ON EARLY STEPS

While standard discriminative models such as DPT Ranftl et al. (2021) struggle to extract reliable control signals from blurry images, prior work has shown that intermediate diffusion features already encode spatial structure information even at early stages of generation Chen et al.. Motivated by this, instead of relying on discriminative models for images, we propose training a lightweight convolutional network $\mathbb{H}(\cdot, t)$ to estimate the control signals directly from intermediate diffusion features at every denoising step. This idea is inspired by Readout Guidance Luo et al. (2024), which trains small timestep-conditioned models to extract signals from diffusion features. To validate the effectiveness of intermediate feature-based signal estimation, we compare the predictions of $\mathbb{H}(\cdot, t)$ with those of the standard DPT depth estimator Ranftl et al. (2021) (Fig. 3). Depth prediction accuracy and control alignment were measured using RMSE between the two approaches: DPT applied to single-step predictions versus $\mathbb{H}(\cdot, t)$ applied to intermediate features.

Our results demonstrate that $\mathbb{H}(\cdot, t)$ predicts results more aligned with the final depth prediction for SD1.5, especially at the early stage of generation Fig. 2. Additionally, $\mathbb{H}(\cdot, t)$ proves to be more stable for ControlNet generation, maintaining consistent signal estimation throughout the entire denoising trajectory (Fig. 2, *Right*). These findings indicate that intermediate features provide more robust signal predictions, enabling accurate control signal estimation even in high-noise regimes. By leveraging $\mathbb{H}(\cdot, t)$ to enforce alignment at early stages of generation, we introduce **InnerControl** – the method that suggests the new training objective that addresses the misalignment that appeared in the previous approaches:

$$\mathcal{L}_{\text{alignment}} = \mathcal{L}\left(c_{\text{spatial}}, H_t\right) \tag{7}$$

where $H_t = \mathbb{H}\left(\text{ControlNet}(c_{\text{spat}}, c_{\text{txt}}, x_T, t), t\right)$ is the lightweight convolutional estimator applied to ControlNet features at timestep $t$.

This loss is applied during training to improve control alignment. The additional alignment block is illustrated in Fig. 4.

For the final training objective, we use a weighted combination of the standard diffusion loss 4, reward loss at the early denoising stage 6, and the additional alignment loss 7:

$$\mathcal{L}_{\text{training}} = \mathcal{L}_{\text{diffusion}} + \alpha \cdot \mathcal{L}_{\text{reward}} + \beta \cdot \mathcal{L}_{\text{alignment}} \tag{8}$$

This additional loss penalizes discrepancies between $c_{spatial}$ and $\hat{c}_{spatial}$ at each timestep $t$, enforcing spatial alignment throughout the denoising process and resulting in improved control correspondence and image fidelity, without visible artifacts compared to reward loss at early steps (Fig. 1).

# 5 EXPERIMENTS

## 5.1 EXPERIMENT SETUP

**Datasets.** Our method was evaluated on multiple datasets, corresponding to a specific task. We evaluated LineArt and HED conditioning on the MultiGen-20M dataset Qin et al. (2023), a large-scale synthetic dataset containing paired images and control signals. For depth estimation, we utilized the corresponding MultiGen-20M depth dataset, which provides precomputed depth maps generated using standard monocular depth estimation techniques. Segmentation experiments were conducted on ADE20K Zhou et al. (2019; 2017).

**Implementation details.** To ensure a fair comparison, we trained our model, ControlNet++ Li et al. (2024) and Ctrl-U Zhang et al. (2024) under identical experimental settings. For each type of control, we first finetuned the pretrained ControlNet model using the AdamW optimizer with a learning rate of $10^{-5}$. This stage runs for 5k iterations for segmentation and 10k iterations for all other tasks. After finetuning, we continue training for another 5k iterations for segmentation and 10k iterations for the remaining tasks, using the same optimization settings and the proposed loss 8. Specifically, the proposed $\mathcal{L}_{\text{alignment}}$ loss was applied over different ranges of diffusion steps: $[920, 0]$ for depth, $[800, 0]$ for HED, $[700, 0]$ for LineArt, and $[980, 450]$ for segmentation. The reward loss was applied over 200 steps for the LineArt and segmentation tasks, and over 400 steps for the depth and HED tasks. All experiments used $512 \times 512$ images with a batch size of 256. See the supplementary material for detailed model settings.

**Baselines.** We compare our method against several competitors, including T2I-Adapter Mou et al. (2024), ControlNet v1.1 Zhang et al. (2023), GLIGEN Li et al. (2023b), Uni-ControlNet Zhao et al. (2023) and and UniControl Qin et al. (2023) and ControlNet++ Li et al. (2024) and CTRL-U Zhang et al. (2024). Most of these methods are based on SD1.5 for text-to-image generation, but we additionally include several models based on SDXL Podell et al. (2023): ControlNet-SDXL and T2I-Adapter-SDXL, following the evaluation protocol suggested in CTRL-U Zhang et al. (2024). For a fair comparison, all models were evaluated under identical image conditions and text prompts, using guidance scale 7.5.

**Metrics and evaluation.** We evaluate alignment fidelity using task-specific metrics: Structural Similarity Index (SSIM) between generated edges and input control signals, Root Mean Squared Error (RMSE) between predicted and ground-truth depth maps and mIoU for the segmentation task. All metrics were computed on the $512 \times 512$ images to ensure consistency. To reduce stochastic variance, we generated 4 independent sample batches with different random seeds and reported the mean metrics. More information about evaluation models can be found in the supplementary material.

## 5.2 EXPERIMENTAL RESULTS

**Comparison of Controllability.** We summarize the results on control alignment quality in Table 1. Our method achieves notable improvements over the baselines for both depth and segmentation estimation: RMSE is reduced by $5.6\%$ and the mean Intersection over Union (mIoU) is improved by $5.6\%$ compared to ControlNet++ Li et al. (2024). These results highlight stronger alignment with the control signals, particularly under high guidance intensity. Furthermore, for edge-based control tasks (LineArt and HED), our approach outperforms both ControlNet++ and CTRL-U in terms of SSIM.

**Comparison of Image Quality.** To evaluate the perceptual quality of generated images, we report the Fréchet Inception Distance (FID) Heusel et al. (2017) for all evaluated methods at a guidance scale of 7.5. As shown in Table 1, our method achieves the best controllability metrics without sacrificing FID. Moreover, for both depth and segmentation control tasks InnerContol improves the image quality of ControlNet++ Li et al. (2024). While our FID scores are not as strong as CTRL-U for the LineArt and segmentation tasks, we obtain substantially higher controllability, outperforming it by 5% on LineArt and 9% on segmentation. Finally, we note that our alignment loss could also be integrated into the CTRL-U Zhang et al. (2024) pipeline, which represents a promising direction for future work.

| Method | T2I Model | Hed Edge | | LineArt Edge | | Depth Map | | Segmentation | |
|---|---|---|---|---|---|---|---|---|---|
| | | SSIM ↑ | FID ↓ | SSIM ↑ | FID ↓ | RMSE ↓ | FID ↓ | mIoU ↑ | FID ↓ |
| *Guidance scale = 7.5* | | | | | | | | | |
| ControlNet | SDXL | — | — | — | — | 40.00 | — | — | — |
| T2I-Adapter | SDXL | — | — | 0.639 | — | 39.75 | — | — | — |
| T2I-Adapter | SD1.5 | — | — | — | — | 48.40 | 22.52 | — | — |
| Gligen | SD1.4 | 0.563 | — | — | — | 38.83 | 18.36 | — | — |
| Uni-ControlNet | SD1.5 | 0.691 | 17.1 | — | — | 40.65 | 20.27 | — | — |
| UniControl | SD1.5 | 0.797 | 16.0 | — | — | 39.18 | 18.66 | — | — |
| ControlNet | SD1.5 | 0.762 | 15.4 | 0.705 | 17.4 | 35.90 | **17.76** | 32.60 | 41.1 |
| ControlNet++ | SD1.5 | 0.822 | **13.0** | 0.840 | 13.2 | 27.63 | 18.59 | 38.08 | 39.04 |
| Ctrl-U | SD1.5 | 0.820 | 13.2 | 0.810 | **12.5** | 26.50 | 18.67 | 36.95 | **35.00** |
| **InnerControl** (Ours) | SD1.5 | **0.826** | 13.0 | **0.850** | 13.5 | **26.09** | 18.29 | **40.22** | 37.65 |

Table 1: Unified comparison on the MultiGen-20M benchmark. Controllability is evaluated using SSIM (↑) for HED/LineArt, RMSE (↓) for depth, and mIoU (↑) for segmentation; fidelity is measured by FID (↓)

**Qualitative Analysis.** We present a qualitative comparison of image generations, showing side-by-side results from our method, ControlNet Zhang et al. (2023), ControlNet++ Li et al. (2024), and CTRL-U Zhang et al. (2024) under identical prompts and control signals (i.e., depth maps, segmentation maps, HED edges, and LineArt edges) at a guidance scale of 7.5. As highlighted in Figure 5, we observed misalignments between the input conditions and the generated results of the competing models. Specifically, ControlNet produced noisy edges for both LineArt and HED control tasks, while ControlNet++ and CTRL-U generated images with noticeable artifacts and inconsistent object distances when using depth control. Furthermore, all competitor models exhibited inaccurate object placement relative to the input segmentation mask during the segmentation task.

## 5.3 ABLATION

**Alignment steps ablation.** We conducted an ablation study to analyze how the application of alignment and reward losses across different diffusion denoising steps affects performance (Table 6). Specifically, we trained ControlNet models with alignment ($\mathcal{L}_{\text{alignment}}$) and reward ($\mathcal{L}_{\text{reward}}$) losses applied to varying subsets of denoising steps. All models were initialized from open-source ControlNet weights for depth estimation and trained under identical conditions (same seed, iteration count, and optimizer settings). Our experiments show that integrating alignment loss into both ControlNet and ControlNet++ training pipelines improves control alignment (RMSE) and image quality (FID). However, using alignment loss alone is less effective than reward loss.

We further investigated the impact of extending the alignment loss to different numbers of denoising steps. Unlike reward loss, applying alignment loss to early steps does not increase FID. For this reason, in our main experiments, we apply alignment loss across 920 denoising steps.

| Method | Steps | Steps Reward | RMSE ↓ | FID ↓ | CLIP ↑ |
|---|---|---|---|---|---|
| *Guidance scale = 7.5* | | | | | |
| ControlNet | 0 | 0 | 33.95 | 18.61 | **32.175** |
| ControlNet | 920 | 0 | 32.80 | 18.55 | 32.05 |
| ControlNet | 0 | 920 | **25.70** | 22.43 | 31.43 |
| ControlNet | 0 | 200 | 29.66 | 18.51 | 32.11 |
| ControlNet | 200 | 200 | 28.93 | 18.32 | 32.06 |
| ControlNet | 400 | 200 | 28.56 | 18.57 | 32.00 |
| ControlNet | 600 | 200 | 28.41 | 18.52 | 31.99 |
| ControlNet | 920 | 200 | 27.50 | **18.22** | 31.92 |

Table 2: Ablation study on the Depth Map control task from the MultiGen-20M benchmark, analyzing the influence of the number of timesteps over which reward and alignment losses are applied.

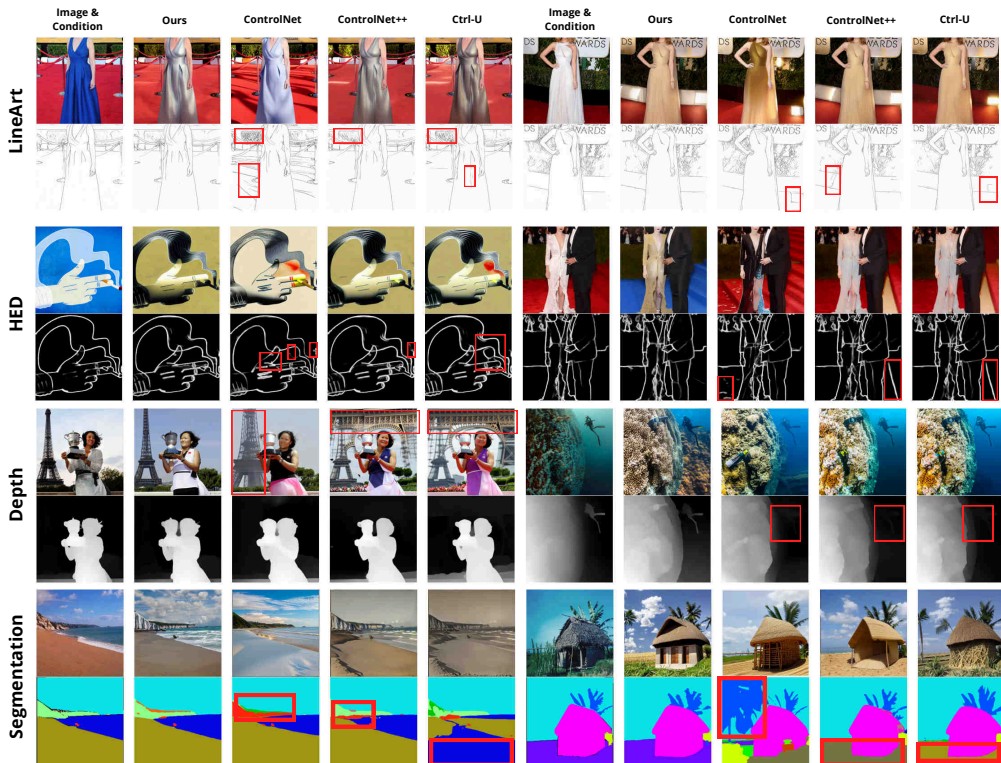

Figure 5: **Qualitative Comparison with Baselines:** Side-by-side results for LineArt (top), HED and Depth (middle) and segmentation control (bottom) using identical prompts and a guidance scale of 7.5. Our method produces results that are more accurate and better aligned with the input controls compared to competing approaches.

## 6  CONCLUSION

In this work, we address the challenge of improving ControlNet controllability by refining its training objective to enforce consistency between the input controls and intermediate diffusion features. We analyze the limitations of previous reward-based approaches, ControlNet++Li et al. (2024) and CTRL-U Zhang et al. (2024), which focus on control alignment during the final diffusion steps while neglecting early denoising stages, where spatial structure predominantly emerges Chen et al.. To overcome this limitation, we propose improved training strategy, that utilizes a lightweight convolutional network that extracts control signals from intermediate features at every diffusion step. This allows us to enforce an explicit alignment loss across the entire sampling trajectory. We conducted experiments on four distinct control benchmarks – LineART, HED, segmentation, and depth map control. Our results consistently demonstrate that this alignment strategy significantly improves controllability. Crucially, it achieves this without decreasing image quality, and in the case of depth and segmentation maps, it actually leads to an improvement in image quality. These results highlight the importance of enforcing consistency throughout the entire diffusion trajectory and underscore the great potential of our approach for future studies.

### 6.1  REPRODUCIBILITY STATEMENT.

For complete reproducibility, we conducted our training using fixed seeds across all competitors. We provide detailed information regarding the training and evaluation procedures A.1, 5.1, including the optimizer used, the number of steps, and the specific hyperparameters chosen for training our models A.1. Since our experiments rely on publicly available datasets, all reported results can be fully reproduced.

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

# A APPENDIX

## A.1 IMPLEMENTATION DETAILS

**Training details.** We utilize the same $\alpha$ for the reward loss weights as suggested in ControlNet++: 0.5 for the depth and segmentation control, 1 for the HED and 10 for LineArt control with a timestep threshold of 200 steps for LineArt and segmentation tasks and 400 steps for depth and HED control. For the alignment loss, training details are presented in the Table 5. Datasets information is shown in Table 4. The training was conducted on $8\ H100$ GPUs and took around 6 hours.

We additionally estimate throughput and memory usage for ControlNet++ and our pipeline for 4H100 with batch size 8. The results are presented in Table 3

| Component | Time per iteration | Peak Memory Delta per GPU |
|---|---|---|
| Reward-only step | 0.15 s | - |
| Alignment-only step | 0.01 s | - |
| Full training iteration (rewarding) | 0.47 s | 41 Gb |
| Full training iteration (ours) | 0.46 s | 44 Gb |

Table 3: Component Performance Metrics

Our codebase is based on the implementation in HuggingFace's Diffusers Von Platen et al. (2022).

**Reward models details.** We additionally provide information about reward models in Table 5. Following ControlNet++ Li et al. (2024) and CTRL-U Zhang et al. (2024) we utilize a slightly weaker model as the reward model for depth estimation and segmentation control training and a stronger model for evaluation. For HED and LineArt we use the same models as proposed in ControlNet Zhang et al. (2023).

| | Segmentation mask | HED Edge | LineArt Edge | Depth Map |
|---|---|---|---|---|
| Dataset | ADE20K Zhou et al. (2019; 2017) | MultiGen20M Qin et al. (2023) | MultiGen20M Qin et al. (2023) | MultiGen20M Qin et al. (2023) |
| Training Samples | 20,210 | 2,560,000 | 2,560,000 | 2,560,000 |
| Evaluation Samples | 2,000 | 5,000 | 5,000 | 5,000 |
| Evaluation Metric | mIoU $\uparrow$ | SSIM $\uparrow$ | SSIM $\uparrow$ | RMSE $\downarrow$ |

Table 4: Datasets and evaluation details for explored tasks. $\uparrow$ denotes higher is better, $\downarrow$ – lower is better.

| | Segmentation Mask | Depth Edge | Hed Edge | LineArt Edge |
|---|---|---|---|---|
| Reward Model (RM) | UperNet-R50 | DPT-Hybrid | ControlNet* | ControlNet* |
| RM Performance | ADE20K(mIoU): 42.05 | NYU(AbsRel): 8.69 | - | - |
| Evaluation Model (EM) | Mask2Former | DPT-Large | ControlNet* | ControlNet* |
| EM Performance | ADE20K(mIoU): 56.01 | NYU(AbsRel): 8.32 | - | - |
| Reward Loss Loss Weight $\alpha$ Steps threshold | CrossEntropy Loss 0.5 200 | MSE Loss 0.5 400 | MSE Loss 1.0 400 | MSE Loss 10.0 200 |
| Alignment Loss Loss Weight $\beta$ Steps threshold | CrossEntropy Loss 0.05 [450, 980] | MSE Loss 1.0 [0, 920] | Sparse MSE Loss 0.1 [0, 800] | MSE Loss 2.0 [0, 700] |

Table 5: Details about some training parameters and reward models. ControlNet* denotes utilizing the same model to extract signal as ControlNet Zhang et al. (2023)

**Additional ablations.** We conducted additional experiments to explore the impact of the alignment loss coefficient $\beta$ and the specific timesteps during which the alignment loss is applied for the segmentation task. The results of this ablation study are presented in Tables 7 and 6. The data indicates

that the segmentation model is more efficient and performs better when the alignment loss is applied during the second part of the diffusion generation timesteps.

**Timesteps schedules.** Intervals for alignment loss application were determined through empirical grid search. However, rather than blindly searching over all possible intervals, we first evaluated the probe's prediction quality (e.g., cross-entropy for segmentation, MSE for depth) across diffusion timesteps and compared it against the full reward model's signal. For segmentation, we observed that probe predictions become unreliable in the final denoising steps Figure 8, which motivated us to exclude these late timesteps from the alignment loss. Guided by this observation, we then performed a coarse grid search over start/end points to identify the interval that maximized controllability without degrading image quality.

| $\beta$ | loss | g.s. | RMSE | FID | CLIP |
|---|---|---|---|---|---|
| 0.5 | MSE | 7.5 | 26.09 | 17.82 | 31.72 |
| 1 | MSE | 7.5 | 26.09 | 18.29 | 32.00 |
| 2 | MSE | 7.5 | 26.21 | 18.24 | 31.82 |
| 5 | MSE | 7.5 | 25.98 | 18.08 | 31.68 |

Table 6: Ablations on different loss $\beta$ for alignment loss for depth control task.

| $\beta$ | steps | loss | g.s. | mIoU | FID | CLIP |
|---|---|---|---|---|---|---|
| 0.05 | [400, 980] | CrossEntropy | 7.5 | 40.22 | 37.7 | 30.2 |
| 0.1 | [450, 980] | CrossEntropy | 7.5 | 41.68 | 40.9 | 29.90 |
| 0.1 | [400, 980] | CrossEntropy | 7.5 | 41.68 | 43.0 | 29.91 |
| 0.1 | [450, 920] | CrossEntropy | 7.5 | 40.20 | 41.1 | 26.25 |

Table 7: Ablations on different $\beta$ and steps for for alignment loss for segmentation control task.

**Comparison of CLIP Score.** To estimate prompt alignment, we calculate CLIP-Score metrics, providing the results in Table 8. We calculate CLIP metrics for ControlNet++ and CTRL-U using our trained versions. We observe that while providing more aligned and quality images, we remain at the same CLIP score level in various setups.

| Method | T2I Model | Hed CLIP ↑ | LineArt CLIP ↑ | Depth CLIP ↑ | Segmentation CLIP ↑ |
|---|---|---|---|---|---|
| *Guidance scale = 7.5* | | | | | |
| T2I-Adapter | SD1.5 | — | — | 31.46 | — |
| Gligen | SD1.4 | — | — | 31.48 | — |
| Uni-ControlNet | SD1.5 | 31.94 | — | 31.66 | — |
| UniControl | SD1.5 | 32.02 | — | 31.68 | — |
| ControlNet | SD1.5 | 31.46 | 31.26 | 32.05 | 30.6 |
| ControlNet++ | SD1.5 | 32.05 | 31.87 | 32.0 | 30.9 |
| Ctrl-U | SD1.5 | 32.05 | 31.88 | 31.9 | 31.2 |
| **InnerControl** (Ours) | SD1.5 | 32.05 | 31.78 | 32.0 | 30.2 |

Table 8: Per-condition semantic alignment measured by CLIP-score (↑).

**Alignment models details.** For our work, we utilize the architecture for $\mathbb{H}(\cdot, t)$ from ReadoutGuidance Luo et al. (2024). Following Luo et al. (2024), we build an aggregation network that takes features from the UNet decoder, and applies bottleneck layers He et al. (2016) to standardize the channel count and aggregate with a learned weighted sum. Additionally, these models use pre-trained timesteps embedding for model conditioning to make predictions on each diffusion step. In order to achieve better results, we add slight modifications to model's architecture. For the depth control task, where we utilize the self-attention features from the UNet decoder instead of convolutional features, as it provides slight improvements in MSE metrics (see Figure 7). To improve edge control, such as HED Edge and LineArt Edge, we resize all features from the UNet decoder to a size of 128 instead of the original 64 and add a 2D transposed convolution operator in the

output head to upsample the image. With these modifications, thin lines in this group of controls are extracted more accurately. For segmentation maps used as input controls, we employ our main baseline model, as it already demonstrates strong performance. These models contains around 8.5M parameters. For comparison, Stable Diffusion 1.5 has  980M parameters, meaning our probes are roughly 100x smaller than the main model.

The models were trained for 10000 steps for edge tasks and 5000 for depth and segmentation using the Adam optimizer with a batch size of 8, while the learning rate was kept fixed at 1e-3 throughout training. For training, we employed several datasets: ADE20K Zhou et al. (2019; 2017) for the segmentation task, PascalVOC Everingham et al. (2010) to annotate custom controls for HED Edges with the HED model and LineArt Edges using extractors proposed in ControlNet Zhang et al. (2023)), and DPT Ranftl et al. (2021) for depth estimation. The approximate training time of a single model on an NVIDIA V100 GPU was about 2.5 hours. The main loss function used across all models was mean squared error (MSE), except for the segmentation task, where cross-entropy loss was applied.

For a more detailed look, we provide additional visualizations of the intermediate control signals, including the extracted depth, segmentation maps, and edges (HED and LineArt), in Figs 9, 11, 10, and 12. Additionally we calculate Cross-Entropy (CE) for segmentation and MSE for edge tasks to compare reward modeling and signal from intermediate features 8. These figures reveal that standard discriminative models struggle to extract precise control information from blurred images, especially for depth and segmentation estimation tasks.

**Unified alignment model.** While training separate models is still doesn't require a lot of efforts, to directly address the concern about overall pipeline complexity, we also trained a single multi-task probe capable of handling depth, HED, and LineArt jointly. This unified model has 9.49M parameters, uses a shared bottleneck with task-specific heads, and is trained by randomly sampling the task at each iteration and applying the corresponding supervised loss. Because the model learns three tasks simultaneously, training was extended to 30,000 iterations. This model achieves performance results comparable to those of separate, individual models. The evaluation was conducted using UNet generaiton on the PascalVOC dataset.

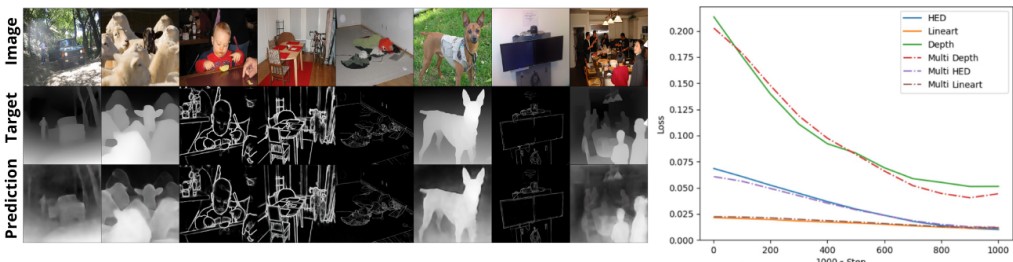

Figure 6: Quantitative and Qualitative Comparison of Models. Results comparing the performance of the unified model against individual, single-task models on three diverse visual tasks: Depth estimation, Holistically-Nested Edge Detection (HED), and LineArt extraction.

**Intermediate features.** We compared extracted feature alignment with the control for the depth estimation task, Fig 13. The top row illustrates that after ControlNet training, extracted depth maps exhibit high correspondence with the input control signal across different steps. This visualization proves the efficiency of alignment across the sampling trajectory, improving alignment not only of extracted features but also the resulting generated image.

## A.2 DISCUSSION

### Alternative to one-step prediction
While one-step prediction often produces noisy results on the early steps of generation, utilizing more sophisticated samplers like DDIM typically incurs high time and memory costs. We significantly optimized the DDIM sampling process by applying gradient checkpointing, and were able to run the full 50-step DDIM sampling with a peak memory usage of 66,265 MB (∼66 GB) and a batch size 1 processing time of ∼3.83 seconds per loss computation. When reducing the number of

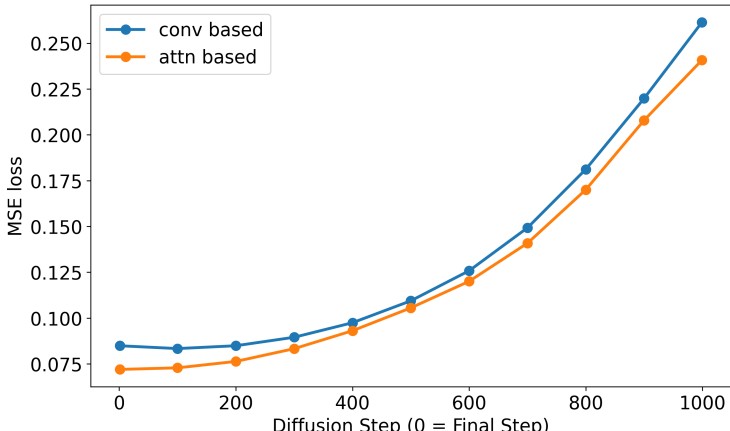

Figure 7: Quality comparison for attention-based and convolution-based predictions for depth maps extraction task

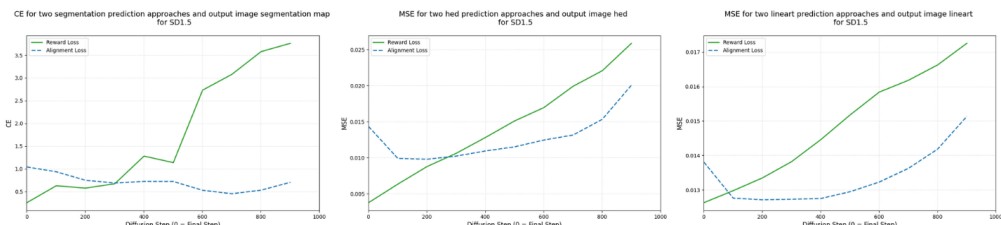

Figure 8: Comparison of signal estimated from one-step prediction and intermediate features for Segmentation, Hed and LineArt control tasks.

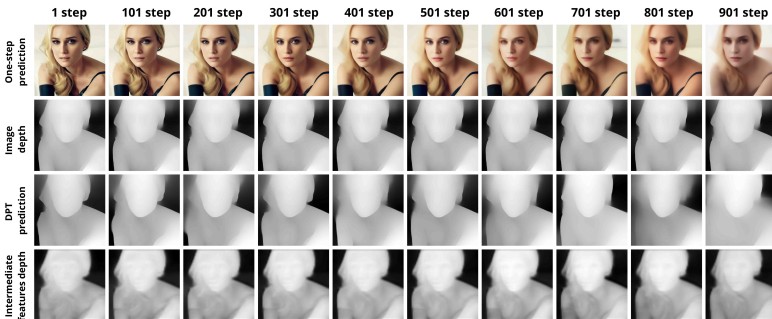

Figure 9: Visualization of one-step prediction, estimated depth of the generated image, corresponding DPT depth estimation, and depth extracted from intermediate features.

steps to 25, we can fit a batch size of 2 on a single H100, with a peak memory of 73,298 MB ($\sim$73 GB) and a slightly higher per-batch time of $\sim$3.98 seconds. Reducing sampling steps more again leads to poor quality, see Figure 14

**Utilizing small networks during inference**

The idea of using the probe at inference time for on-the-fly correction is closely related to Readout Guidance Luo et al. (2024), which proposes augmenting Classifier-Free Guidance with an additional gradient for on-the-fly correction:

$$\hat{\epsilon}_t \leftarrow \epsilon(x_t) + w \cdot \nabla_{x_t} d(r, f(x_t)) \tag{9}$$

where $f(\cdot)$ is the small prediction network (analogous to our probe), $r$ is the target control map, and $d(\cdot, \cdot)$ is a distance function. We evaluated this approach using hed control. While theoret-

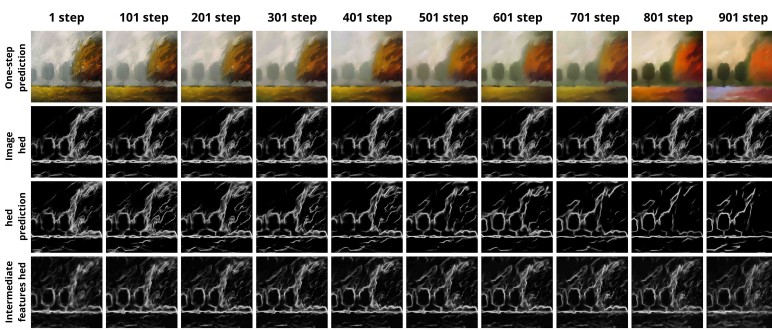

Figure 10: Visualization of one-step prediction, estimated HED of the generated image, corresponding HED estimation, and HED extracted from intermediate features.

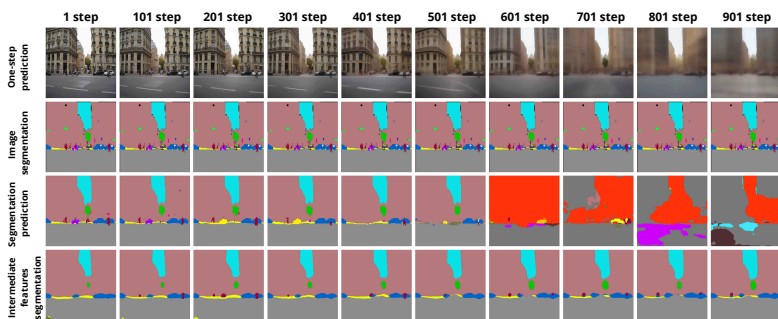

Figure 11: Visualization of one-step prediction, estimated segmentation mask of the generated image, corresponding segmentation maps, and segmentation maps extracted from intermediate features.

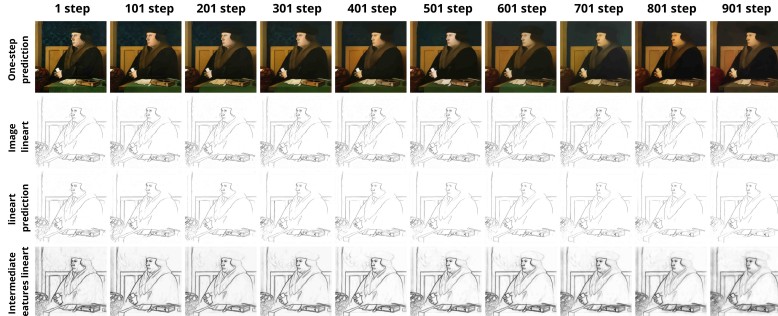

Figure 12: Visualization of one-step prediction, estimated LineArt of the generated image, corresponding LineArt estimation, and LineArt extracted from intermediate features.

ically appealing, this approach failed to improve controllability and severely degraded efficiency. The generated images exhibited poor alignment with the input control (e.g., HED), often failing to incorporate essential structural cues Figure 15.

Another important limitation of such approach is inference time increasing. The on-the-fly gradient calculation, even when applied only during the first half of sampling, caused the inference time to increase. A single 50-step DDIM sample with SD 1.5 inflated from approximately 30 seconds to 1 minute 47 seconds. Thus, our findings demonstrate that the lightweight prediction networks are highly effective when integrated into the training-time alignment loss (as shown by our core results), applying them for on-the-fly inference correction degrades efficiency and fails to improve output controllability.

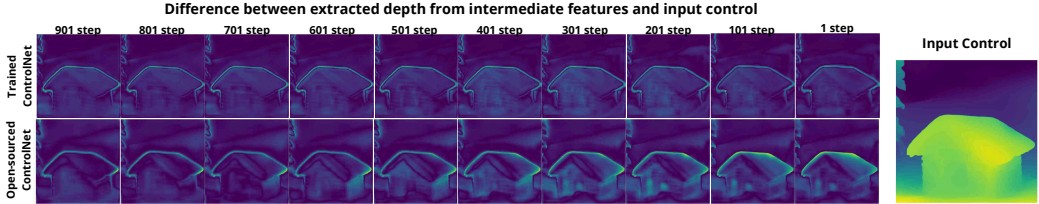

Figure 13: Visualization of difference between extracted signal from intermediate features and input control after our training applied (*top*) and for standard ControlNet (*bottom*)

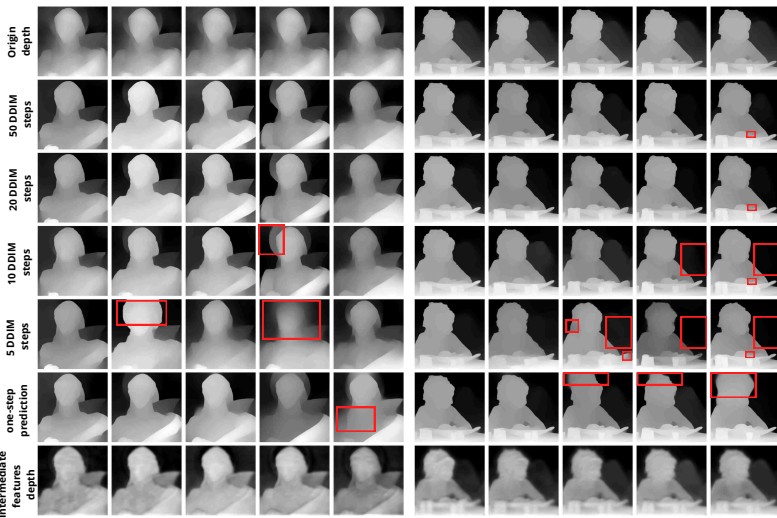

Figure 14: Visualization of different number of sampling steps for depth prediction.

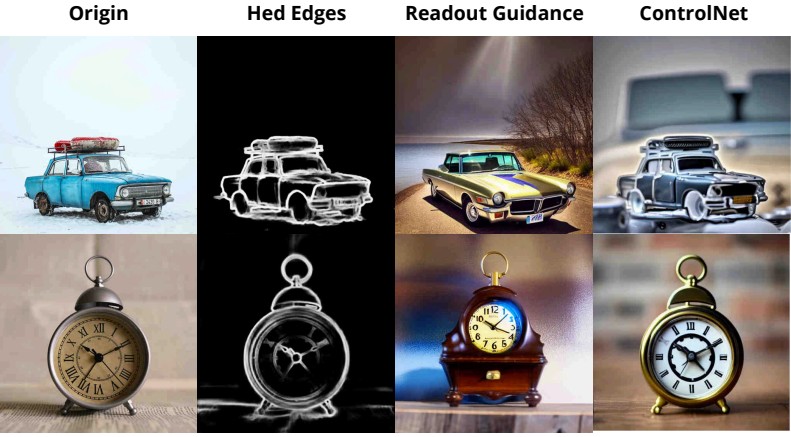

Figure 15: Application small networks for guidance during inference.

## A.3 LIMITATIONS.

The main limitation of our approach is the quality of small convolutional neural nets for signal estimation from intermediate features. Due to their small parameter count and shallow design, these models may struggle to predict fine-grained spatial details, such as thin edges. However, we emphasize that this limitation is not intrinsic to the method itself. Our framework may utilize any model

capable of extracting a signal at each timestep. This opens a promising direction for developing a better model in future work.

### A.4 ETHIC STATEMENT

Our method, built upon the Stable Diffusion 1.5 model, consequently inherits all its possible problematic biases and limitations.

### A.5 MORE VISUALIZATIONS

We also provide visualizations for different control types for InnerControl generation. The results are shown in Figures 16,17,18.

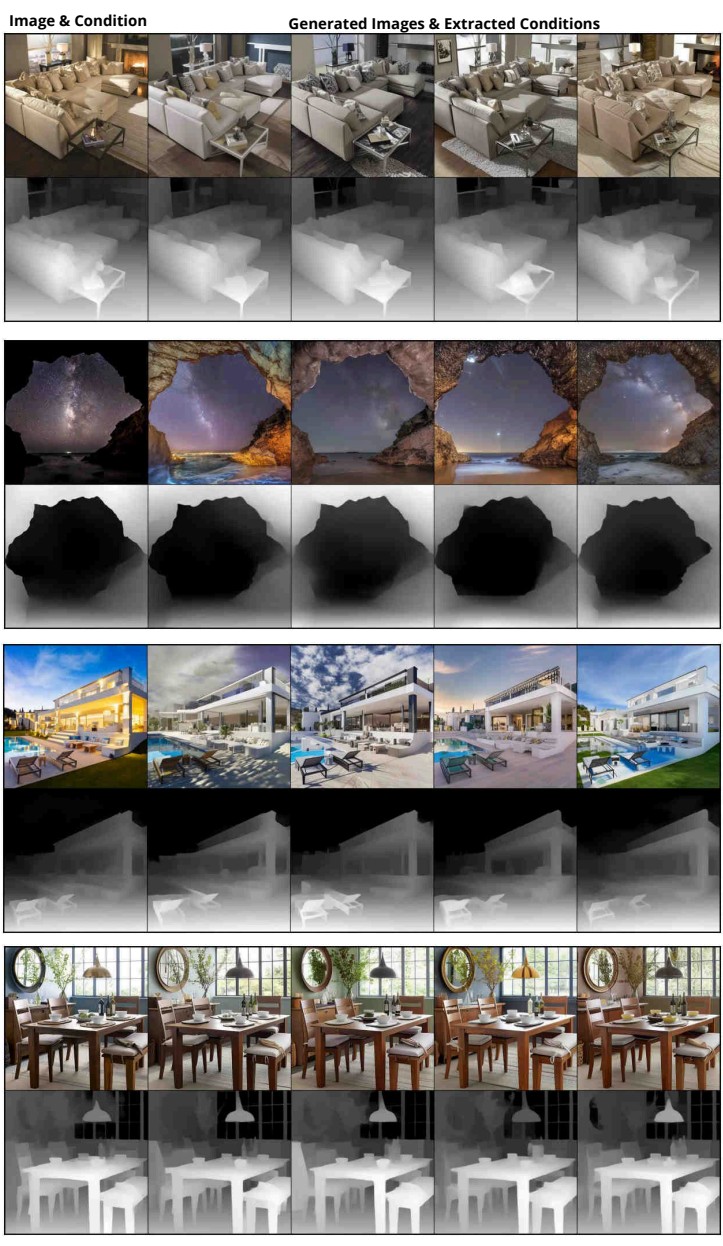

Figure 16: More visualizations for InnerControl (ours) method (depth maps)

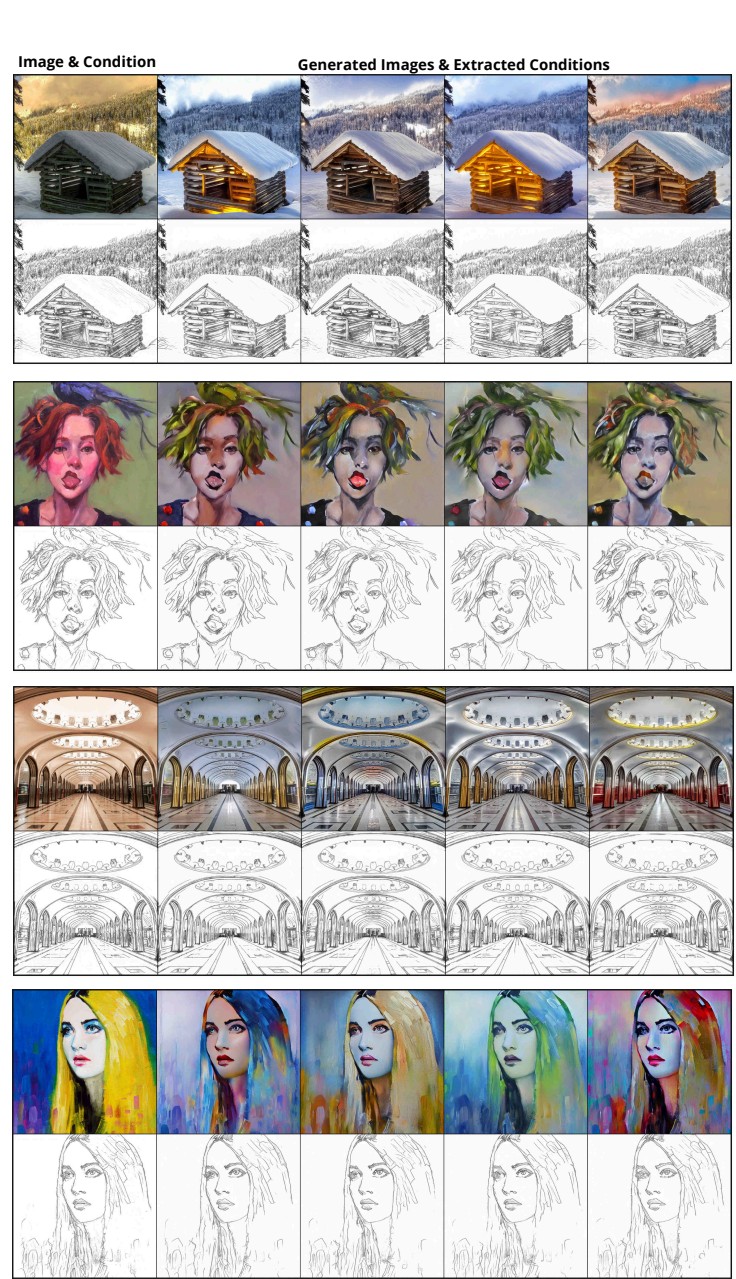

Figure 17: More visualizations for InnerControl (ours) method (LineArt)

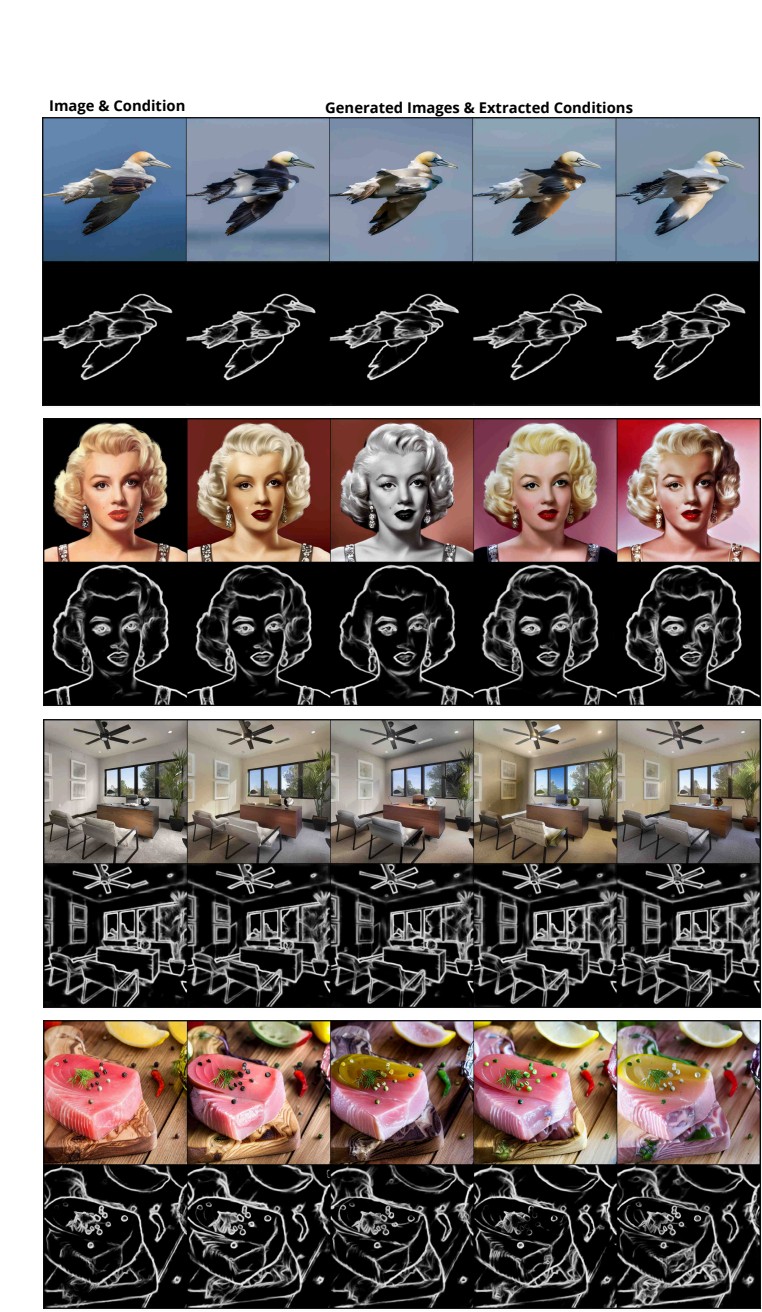

Figure 18: More visualizations for InnerControl (ours) method (HED)

