# OpenReview forum: "Heeding the Inner Voice: Aligning ControlNet Training via Intermediate Features Feedback"
_ICLR.cc/2026/Conference — Submitted to ICLR 2026_

### Official Review · Reviewer_5Wbu · 2025-10-29

**Soundness:** 2
**Presentation:** 2
**Contribution:** 2
**Rating:** 4
**Confidence:** 3

**Summary:**

This paper introduces InnerControl, a novel training strategy to improve spatial alignment in ControlNet-based text-to-image diffusion models. While existing methods like ControlNet++ enhance controllability by applying cycle consistency losses, they primarily focus on late denoising steps, neglecting the early stages where spatial structure begins to form. This limitation leads to misalignments between input control signals (e.g., edge maps, depth, segmentation) and the final generated images, especially when extending supervision to early steps causes visual artifacts.

To address this, the authors propose leveraging intermediate UNet features across all denoising steps, not just the final ones. They train lightweight, timestep-conditioned convolutional probes to predict control signals (e.g., depth, edges) directly from these intermediate features. These probes are used to generate pseudo ground truth controls during training, enabling an alignment loss that enforces consistency between the predicted and target control at every diffusion timestep.

The key insight is that intermediate diffusion features encode spatial information even in noisy early stages, making them more reliable than one-step image predictions for supervision. This allows InnerControl to avoid artifacts while improving control fidelity and image quality.

Main Contributions of this paper can be summarized:

A new training objective that enforces consistency between input controls and intermediate diffusion features throughout the entire denoising process, including early steps.

The proposed method improves upon reward-based approaches like ControlNet++ and CTRL-U, achieving better alignment (e.g., 5.6% lower RMSE for depth, 5.6% higher mIoU for segmentation) without degrading image quality.

The alignment loss can be integrated into existing frameworks, offering a plug-and-play enhancement for various control types (edge, depth, segmentation).

**Strengths:**

The paper’s core contribution—leveraging intermediate diffusion features for control alignment across all denoising steps—is both novel and timely. While prior works like ControlNet++ and CTRL-U focused on late-stage alignment via reward losses, this work identifies and addresses a critical temporal gap: the early denoising stages, where spatial structure emerges but is overlooked. The idea of training lightweight, timestep-conditioned probes to extract control signals from noisy intermediate features is creative and departs significantly from the dominant paradigm of using single-step image predictions for supervision. This approach also draws inspiration from recent work on diffusion features for vision tasks (e.g., Readout Guidance), but applies it in a new context: training-time alignment for controllable generation, not post-hoc analysis or guidance.


This work has broad implications for the controllable generation community. By showing that early-stage supervision is not only possible but beneficial, it challenges a key assumption in prior reward-based methods and opens a new dimension for training diffusion-based controllers. The proposed alignment loss is model-agnostic and can be plugged into existing frameworks, making it immediately useful for practitioners. Furthermore, the paper bridges two previously disconnected lines of work: diffusion feature representations (used for vision tasks) and controllable generation, showing that the former can enhance the latter. This cross-pollination is likely to inspire follow-up work in areas like video generation, multi-modal control, or real-time editing. Finally, by removing the temporal limitation of prior reward losses, InnerControl sets a new standard for fine-grained, training-based control in diffusion models.


The paper conducts thorough empirical validation across multiple control tasks (depth, edge, segmentation) and datasets (MultiGen-20M, ADE20K). The ablation studies are well-designed, showing that the proposed alignment loss improves both control fidelity (RMSE, SSIM, mIoU) and image quality (FID), even when applied to early denoising steps—a setting where prior methods fail due to artifacts. The authors also carefully isolate the contribution of their method by integrating it into existing pipelines (ControlNet++, CTRL-U) and showing consistent improvements, demonstrating modularity and generalizability. The use of timestep-conditioned probes is validated with quantitative comparisons against standard discriminative models (e.g., DPT), showing superior stability and accuracy across the diffusion trajectory.

**Weaknesses:**

The alignment module is essentially a “conv head on U-Net decoder” whose structure and channel-fusion logic are borrowed wholesale from Readout Guidance (Readout Guidance: Learning Control from Diffusion Features).  The only new twist is timestep conditioning, which is already standard in diffusion literature.  Consequently, the architectural contribution feels thin.

All four tested conditions (depth, HED, LineArt, segmentation) are dense, pixel-to-pixel maps.  The paper never tackles sparse or semantic controls—e.g., bounding boxes, key-points, open-vocabulary phrases, or 3-D poses—where mis-alignment is often discrete rather than ℓ₂ error.  The probes may fail when the signal is a set of 10 key-points rather than a 512×512 map.  A concrete fix is to add at least one sparse task (e.g., COCO key-point conditioning) and report PCK or OKS instead of RMSE/SSIM.  If the head must be redesigned (e.g., heat-map → coordinate regression), the authors should discuss how much extra engineering is required and whether the same head still applies.

Table 6 shows that segmentation needs alignment only in the second half of the trajectory, while depth needs all steps.  The paper offers no principle for choosing the interval; practitioners must grid-search per task.  This reintroduces a meta-hyper-parameter that the method claimed to remove.


Every training step now runs the probe and computes an extra loss on every timestep.

**Questions:**

The alignment head is taken almost verbatim from Readout Guidance (RG).  What architectural changes did you introduce beyond RG’s weighted-sum bottleneck + timestep embedding?  If the only delta is the loss function, please state so explicitly.


Did you try the same head on SD-XL or on a DiT-based diffusion model?  If it fails, how do you reconcile the claim of “model-agnostic” alignment?

Table 6 indicates that segmentation needs alignment only in [450,980], whereas depth needs [0,920].  Did you attempt to learn these intervals instead of grid-search?  If learned, describe the algorithm; if not, please concede this hyper-parameter overhead.


All four conditions are dense pixel maps.  Have you tried sparse controls—COCO key-points or bounding boxes—where the probe must output 17 heat-maps or 4 numbers instead of 512×512 maps?


You report 6 h on 8×H100 but not the slow-down factor relative to ControlNet++.  Please provide:
(a) images/sec per GPU for both runs,
(b) peak memory delta, and
(c) whether gradient checkpointing was enabled for the probe.


The probe is discarded at inference.  Did you explore using it for on-the-fly correction?  For example, after 50 DDIM steps, if probe-RMSE >τ, inject an auxiliary gradient and continue sampling.  What τ values work?  Does quality improve or collapse?

You argue that intermediate features are more reliable than one-step x₀-predictions.  Can you provide a minimal toy experiment that quantifies this?  For example, plot mutual information between (a) intermediate features and ground-truth depth, and (b) single-step x₀ and ground-truth depth, across noise levels.

---

> ### Author Response · Authors · 2025-11-26
> **Response to the questions [Part 1/3]**
>
> We sincerely thank the reviewer for the thoughtful and detailed feedback, as well as for highlighting the key strengths of our work -- namely:
>
> 1. the novelty and timeliness of leveraging intermediate diffusion features for early-stage control alignment;
>
> 2. the broader implications for controllable generation through bridging diffusion feature representations with training-time alignment;
>
> 3. the thoroughness and generalizability demonstrated in our empirical evaluation across tasks, datasets, and integration with existing control frameworks.
>
> Below we would like to clarify all questions.
>
> ## **1.  Architectural changes beyond RG**
>
> The core architecture of our alignment head is closely based on the Readout Guidance (RG) design: a lightweight module that processes intermediate UNet features, incorporates timestep embeddings, and uses a weighted-sum bottleneck to produce the control signal. We explicitly build upon this framework.
>
> However, we introduced **task-motivated architectural refinements** to optimize performance for diverse control signals. For **edge-based controls**, we add a learned **2D transposed-convolution layer** at the head’s output to upsample predictions, which improves the rendering of thin, high-resolution edges. For **depth estimation**, instead of relying on convolutional decoder features (as done in RG), we found that utilizing **self-attention outputs** from the UNet yields slightly more accurate depth maps (see Fig. 6 in the Appendix).
>
> Thus, to answer the reviewer’s question: the only differences relative to RG are these small task-specific refinements, the overall architecture and design principles remain unchanged.
>
> ## **2. Model-agnostic alignment (SD-XL and DiT)**
>
> Our alignment framework is **conceptually model-agnostic**: it operates on intermediate features and only requires a lightweight probe predicting the control signal from these features, independent of the underlying diffusion model.
>
> To verify this, we performed experiments on **SD-XL** using the same probe architecture and the same training protocol as for SD 1.5.
>
> On a single H100, the maximum batch size for reward fine-tuning drops from 8 (SD 1.5) to 2 (SD-XL), substantially increasing training time and memory usage. Because ControlNet++ does not provide SD-XL baselines, we also had to re-implement the reward training pipeline from scratch.
> Given these constraints, we ran a focused **depth experiment** on SD-XL for 5000 steps with batch size 128. The results (shown in the **Table 1**) indicate that our alignment loss still improves RMSE while maintaining approximately the same FID, which is consistent with our SD 1.5 findings.
>
> **Table 1.**  ControlNet++ and Our method comparison for SD-XL on depth control task.
> | Model | RMSE ↓ | FID ↓ |
> | :--- | :--- | :--- |
> | **ControlNet++** | 30.24 | 17.00 |
> | **Ours** | 28.31 | 17.10 |
>
> For DiT-based models, the situation is more difficult: transformer architecture differs significantly from CNN-based UNets, so the probe design must be adapted. This requires engineering effort but does not contradict the underlying idea of learning a lightweight mapping from intermediate features to control signals. We view this as promising future work rather than a limitation.
>
> ## **3. Learning vs. grid-searching timestep intervals**
> The intervals in Table 6 (e.g., [450, 980] for segmentation and [0, 920] for depth) were determined via empirical grid search, not learned automatically.
>
> However, we did not perform the grid search blindly. We first **evaluated the probe’s prediction quality at different timesteps** (cross-entropy for segmentation, MSE for depth) and compared it to the reward model’s signal. For segmentation, probe predictions become unreliable in the final denoising steps, motivating the exclusion of the last timesteps.
>
> Based on this insight, we then conducted a **coarse grid search** to identify the interval that maximizes control accuracy without degrading image quality.
>
> Finding optimal schedules without grid search is an interesting direction for future work. We will include a dedicated appendix section, where we outline this intuition and the procedure described above.

---

> > ### Author Response · Authors · 2025-11-26
> > **Response to the questions [Part 2/3]**
> >
> > ## **4. Sparse controls (keypoints, bounding boxes)**
> >
> > Although our work focuses on **dense control signals** (depth, edges, segmentation), we agree that sparse controls such as COCO keypoints or bounding boxes are important.
> >
> > Current reward-based pipelines (ControlNet++ and CTRL-U) **do not support these tasks**. As ControlNet++ notes: “Our reward finetuning leverages a pre-trained ControlNet and a differentiable reward model. Currently, pre-trained ControlNet for object bounding boxes and differentiable reward models for sketches are lacking. In existing pose models, there are nondifferentiable operations such as the NMS and keypoints grouping. We leave the question of how to extend consistency reward to more conditions to future work.”
> >
> > In contrast, our method **does not suffer from these limitations**. The alignment models can, in principle, be adapted to output heatmaps (for pose) or scalars (for boxes) and trained with appropriate losses. However, running these experiments would require training the corresponding models, searching for and preparing a suitable pose-control dataset (which is non-trivial, as datasets often contain text captions without keypoint heatmaps, or keypoints without correct heatmaps), training our small probe, retraining ControlNet, and conducting an additional grid search for experimental parameters.
> >
> > Given these constraints, and the lack of existing reward-based baselines, we focused on dense controls where fair comparison is possible.
> >
> >
> > ## **5. Throughput, memory, and checkpointing**
> >
> > We appreciate the request for these quantitative metrics to ensure a fair comparison with ControlNet++. Our measurements directly address all points you raised.
> > For all measurements we conducted experiment with batch size 8 on 4H100 gpu without gradient accumulation.
> >
> > **Throughput.**
> > Below we report the per-GPU throughput, which reflects the slow-down factor accurately:
> > Component
> > Time per iteration
> >
> > | Component | Time per iteration |
> > | :--- | :--- |
> > | Reward module pass | 0.15 s |
> > |Alignment module pass | 0.01 s |
> > | Full training iteration (rewarding) | 0.46 s |
> > | Full training iteration (ours) | 0.47 s |
> >
> > **Memory usage.**
> >
> > To measure the memory cost of our probes, we computed the change in peak memory ($\Delta$ peak) between the baseline ControlNet forward/backward pass and our full training iteration. All experiments were run on 4×H100 GPUs with an effective batch size of 32 (batch size 8 × 4 gpus).
> >
> > The observed peak memory delta is:
> >
> > | Component | Peak Memory Delta per GPU |
> > | :--- | :--- |
> > | Reward-only iteration | 41 GB |
> > | Our method (full iteration) | 44 GB |
> >
> > **Gradient checkpointing.**
> >
> > Gradient checkpointing was enabled for the main **ControlNet model** during all runs. However, the probes are very small networks (~8.5M parameters; by comparison, SD 1.5 has ~980M parameters, so the probes are ~100× smaller). Because of their negligible memory footprint, gradient checkpointing was not necessary for the probes and was disabled for the probe modules.
> >
> > ## **6. Using the probe for on-the-fly inference correction**
> >
> > The idea of using the probe at inference time for on-the-fly correction is closely related to Readout Guidance [1], which proposes augmenting Classifier-Free Guidance with an additional gradient for on-the-fly correction:
> >
> >
> > $\hat{\epsilon}_t \leftarrow  \epsilon(x_t) + w \cdot \nabla _{x_t} d(r, f(x_t))$
> >
> > where $f(\cdot)$ is the small prediction network, $r$ is the target control map, and $d(\cdot,\cdot)$ is a distance function.
> >
> > We evaluated this approach using hed control. While theoretically appealing, this approach failed to improve controllability and severely degraded efficiency. The generated images exhibited **poor alignment with the input control** (e.g., HED), often failing to incorporate essential structural cues. We will provide the results of generation in Appendix.
> >
> > Another important limitation of such approach is **inference time increasing**. The on-the-fly gradient calculation, even when applied only during the first half of sampling, caused the inference time to increase. A single 50-step DDIM sample with SD 1.5 inflated from approximately 30 seconds to 1 minute 47 seconds.
> >
> > Thus, our findings demonstrate that the lightweight prediction networks are highly effective when integrated into the training-time alignment loss (as shown by our core results), applying them for on-the-fly inference correction degrades efficiency and fails to improve output controllability.

---

> > > ### Author Response · Authors · 2025-11-26
> > > **Response to the questions [Part 3/3]**
> > >
> > > ## **7. Quantifying reliability of intermediate features vs. one-step prediction**
> > >
> > > We directly compared the pixel-based **RMSE** between the ground-truth depth and the estimations derived from two distinct sources across various noise levels:
> > >
> > > - Signal derived from the **one-step prediction**.
> > >
> > > - Signal derived from **intermediate features**.
> > >
> > > We demonstrate that the signal derived from **intermediate features consistently achieves lower RMSE** compared to the one-step prediction across noise levels. This lower error quantitatively confirms our claim that intermediate features are inherently more reliable and less sensitive to noise than the one-step prediction.
> > >
> > > Our qualitative comparisons (Figures 7, 8, 9, 10) further illustrate the visual stability and detail of the signal estimated from intermediate features.
> > >
> > > To substantiate our claim with comprehensive metrics across diverse tasks, we are conducting additional quantitative evaluations:
> > >
> > > - **Segmentation**: using the **Cross-Entropy (CE)** loss.
> > >
> > > - **Edge Control:** using the **MSE** loss.
> > >
> > > The resulting data will be included in the Appendix. Based on these comprehensive results, we observe that for the segmentation task probe predictions become unreliable in the final denoising steps, while producing much better results in early generation steps compared to reward models.
> > >
> > > ### References
> > > [1] Luo G. et al. Readout guidance: Learning control from diffusion features //Proceedings of the IEEE/CVF Conference on Computer Vision and Pattern Recognition. – 2024. – С. 8217-8227.

---

> > > > ### Author Response · Authors · 2025-11-28
> > > > **Response to the questions**
> > > >
> > > > We have updated the Appendix in our paper. New sections now include:
> > > >
> > > > 1. Timestep scheduler: Appendix A1: Timesteps schedules
> > > > 2. Throughput, memory utilization: Appendix A1: Training details
> > > > 3. On-the-fly inference correction: Appendix A2: Utilizing small networks during inference
> > > > 4. Quantifying reliability of intermediate features vs. one-step prediction: Appendix A1: Alignment models details, Figure 8
> > > >
> > > > All additions and updates are highlighted in blue text for ease of review.

---

### Official Review · Reviewer_e6tR · 2025-10-31

**Soundness:** 3
**Presentation:** 3
**Contribution:** 3
**Rating:** 6
**Confidence:** 4

**Summary:**

This paper addresses a key limitation in ControlNet training: the inability to enforce spatial control alignment during early diffusion steps. Prior methods (like ControlNet++) use a reward loss ($\mathcal{L}_{reward}$) that fails in early steps because it relies on inaccurate, blurry single-step predictions ($x_0'$), causing image artifacts.

The proposed method, InnerControl, solves this by changing the signal source. Instead of using the $x_0'$ prediction, it pre-trains lightweight, timestep-conditioned "probes" ($\mathbb{H}(\cdot, t)$) to extract control signals directly from the UNet's intermediate features (the "inner voice").

This extracted signal is stable and accurate even at high noise levels, enabling a new alignment loss that can be applied across the entire diffusion process without degrading image quality. The final training combines the stable, all-step with the strong, late-step. This approach achieves state-of-the-art control fidelity (e.g., 5.6% RMSE drop in depth) while maintaining or improving image quality (FID).

**Strengths:**

1. The paper's core strength is its clear identification and experimental validation of why prior reward losses (like ControlNet++) fail. The analysis showing that the signal source (the blurry one-step $x_0'$ prediction) is the root cause of early-step instability is a critical insight.
2. The idea of using the UNet's "inner voice" (intermediate features) is an elegant solution. The pre-trained probe ($\mathbb{H}$) is shown to be a far more robust signal extractor than standard models (like DPT) operating on $x_0'$.
3. The method is clever in combining the new stable loss with the old strong loss. It doesn't just replace $\mathcal{L}_{reward}$; it complements it. The alignment loss provides a stable baseline control across all steps, while the reward loss provides a strong correction at the end. The ablations (Table 2) clearly show this combination is superior to either loss alone.
4. The method achieves its primary goal: it measurably improves control alignment (SOTA on RMSE, SSIM, mIoU) without the image quality (FID) trade-off that plagued prior methods when applied to early steps.

**Weaknesses:**

1. The proposed method requires a new, non-trivial pre-training stage for the $\mathbb{H}$ probe. This must be done for every control type (depth, segmentation, HED, etc.), and each probe requires its own specific training dataset (e.g., ADE20K for segmentation). This increases the overall pipeline complexity and data requirements. The paper could be strengthened by discussing the generalization of these probes or a more data-efficient way to train them.
2. The final loss function is a complex combination of three losses, each with weights ($\alpha$, $\beta$) and, critically, task-specific timestep schedules (e.g., depth alignment is applied at [0, 920], segmentation at [450, 980]; reward is 400 steps vs. 200). This suggests the method may be difficult to tune for a new, unseen control type. The paper would be more impactful if it provided a clearer methodology or ablation for how these optimal schedules were determined, or if it showed that a single, unified schedule can work well.
3. The results in Table 1 show that while InnerControl consistently wins on controllability (e.g., SSIM), it sometimes loses to CTRL-U on image quality (e.g., FID for LineArt and Segmentation). This suggests the trade-off is not fully resolved. The paper should discuss this trade-off more directly. Is there a "knob" (e.g., the $\beta$ weight) that could balance this, allowing a user to trade a bit of control for better FID?
4. The paper's premise is that the $x_0'$ prediction is unreliable. The chosen solution is to find a new signal source (intermediate features). An alternative, unexplored path would be to improve the $x_0'$ prediction itself. For example, would using a more sophisticated sampler's $x_0'$ prediction (like DDIM) or a multi-step $x_0'$ prediction provide a stable-enough signal for $\mathcal{L}_{reward}$? A small ablation on this could strengthen the paper's claim that intermediate features are the necessary solution.

**Questions:**

None

---

> ### Author Response · Authors · 2025-11-26
> **Response to the questions [Part 1/2]**
>
> We sincerely thank the reviewer for the **highly constructive feedback**. Below we address each point in detail.
>
> ## **1. On the need for a pre-training stage for probe models**
>
> Thank you for raising this point. While our method does introduce a pre-training stage for the probe models, these probes are extremely lightweight:
>
> - **Edge probe:** 8.50M parameters
>
> - **Depth / Segmentation probe:** 8.49M parameters
>
> For comparison, **Stable Diffusion 1.5 has ~980M parameters**, meaning our probes are roughly **100x smaller** than the main model. As a result, each probe can be trained on a small dataset within just a few hours on a single V100 GPU, and the computational cost of probe training is negligible relative to the main diffusion fine-tuning.
>
> To directly address the concern about overall pipeline complexity, we also trained a **single multi-task probe** capable of handling **depth, HED, and LineArt jointly**. This unified model has **9.49M parameters**, uses a shared bottleneck with task-specific heads, and is trained by randomly sampling the task at each iteration and applying the corresponding supervised loss. Because the model learns three tasks simultaneously, training was extended to **30,000 iterations** (vs. 10,000 for individual probes).
>
> As we will report in the appendix, this unified probe achieves performance **nearly identical** to the individual single-task probes -- both quantitatively (metrics) and qualitatively (generated examples).
>
> Thus, while we use task-specific probes in the main experiments for maximal clarity, the framework does not intrinsically require separate models per control type, and it can generalize well across multiple tasks within a single lightweight probe.
>
> We will include these results in the appendix.
>
> ## **2. On the complexity of the final loss and the timestep schedules**
>
> We appreciate the reviewer’s observation.
>
> The intervals reported in Table 6 (e.g., [450, 980] for segmentation vs. [0, 920] for depth) were indeed determined through empirical grid search. However, rather than blindly searching over all possible intervals, we first evaluated the probe’s prediction quality (e.g., cross-entropy for segmentation, MSE for depth) across diffusion timesteps and compared it against the full reward model’s signal. For segmentation, we observed that probe predictions become unreliable in the final denoising steps, which motivated us to exclude these late timesteps from the alignment loss. Guided by this observation, we then performed a coarse grid search over start/end points to identify the interval that maximized controllability without degrading image quality.
>
> In parallel, we examined the reward model’s behavior. As shown in Figures 2 and 3, depth estimation exhibits noticeable artifacts after approximately 400 reward steps. This provided a justification for extending the reward window from 200 to 400 steps.
> Finding optimal schedules without grid search is an interesting direction for future work. We will include a dedicated appendix section, where we outline this intuition and the procedure described above.

---

> > ### Author Response · Authors · 2025-11-26
> > **Response to the questions [Part 2/2]**
> >
> > ## **3. On the control–quality trade-off**
> >
> > As shown in our ablations (Tables 5-6), increasing β in the alignment loss indeed strengthens control correspondence, although in some cases it may introduce a slight reduction in image quality. Our primary objective, however, is to **improve controllability while maintaining high visual quality**, and this balance depends not only on β but also on the underlying training pipeline.
> > While our main experiments are built on top of **ControlNet++**, the alignment loss is fully plug-and-play and can be integrated into different training pipelines. For example, in Table 2 of the main paper we show that applying our method directly to **ControlNet** improves control metrics without increasing FID (33.95 RMSE, 18.61 FID → 32.80 RMSE, 18.55 FID). We further **train CTRL-U** to demonstrate that adding our alignment loss also improves depth controllability without degrading image quality (26.50 RMSE, 18.67 FID → 25.40 RMSE, 18.56 FID).
> >
> > These results collectively indicate that the control-quality trade-off is influenced by the choice of base model. With an appropriate backbone, our method can improve controllability while retaining or even improving FID.
> >
> > ## **4. On alternative approaches for improving prediction**
> >
> > Thank you for raising this point regarding memory and computational cost. As noted in ControlNet++: “When the batch size is 1 with FP16 mixed precision, the GPU memory required for a single denoising step – and for storing all training gradients – is approximately 6.8 GB. If we use the 50-step inference with the DDIM scheduler, approximately 340 GB of memory would be needed to perform reward fine-tuning on a single sample, which is nearly impossible with current hardware.”
> >
> > In our experiments, by applying gradient checkpointing, we are able to run the full **50-step DDIM sampling** with a peak memory usage of **66,265 MB** ($\sim$ 66 GB) and a batch size 1 processing time of $\sim$**3.83 seconds** per loss computation. When reducing the number of steps to **25**, we can fit a batch size of 2 on a single H100, with a peak memory of **73,298 MB** ($\sim$73 GB) and a slightly higher per-batch time of $\sim$**3.98 seconds**.
> >
> > Moreover, our lightweight probe networks contain only $\sim$**8.5M** parameters and require just around 10 milliseconds per prediction, adding negligible computational overhead and minimal memory usage.
> >
> > These results demonstrate that even “more sophisticated” samplers such as DDIM remain prohibitively expensive in both memory and time for small models like SD 1.5. We agree that adding a dedicated ablation study on memory and runtime would strengthen the paper, and we will include this in the appendix.

---

> > > ### Author Response · Authors · 2025-11-28
> > > **Response to the questions**
> > >
> > > We have updated the Appendix in our paper. New sections now include:
> > >
> > > 1. Information about unified model in Appendix A1: Unified alignment model
> > > 2. Timestep scheduler: Appendix A1: Timesteps schedules
> > > 3. Exploring DDIM samplers: Appendix A2:  Alternative to one-step prediction
> > >
> > > All additions and updates are highlighted in blue text for ease of review.

---

### Official Review · Reviewer_PrLh · 2025-11-01

**Soundness:** 2
**Presentation:** 2
**Contribution:** 2
**Rating:** 2
**Confidence:** 4

**Summary:**

This paper proposing a new training strategy called InnerControl, which is an enhancement to existing ControlNet-style architectures for text-to-image diffusion models that improves spatial alignment between conditioning inputs (like edge maps, depth, or segmentation) and generated images.
* The key idea is to get consistency signal feedback during whole generation process.

**Strengths:**

- This paper observes that prior approaches only focus on the final generation results, leading to slow and delayed feedback during training.
- To address this, the paper introduces the prediction of a pseudo \\( x_0 \\), which is then decoded into an image via a VAE. This enables the extension of the consistency loss to every diffusion step.

**Weaknesses:**

* the cost is expensive because we have to do the vae decode for each step.

* the contribution compared with controlnet++ is only the step level feedback, which is a trival trick since 2023.

* the improvement is marginal.

**Questions:**

* In Table 1, for FID of Depth Map, it seems ControlNet has best performance.

---

> ### Author Response · Authors · 2025-11-26
> **Response to the questions [Part 1/1]**
>
> We thank the reviewer for the insightful feedback and for highlighting an important strength of our work: the observation that prior approaches rely exclusively on the final generated outputs, leading to slow and delayed feedback during training. We appreciate that this key motivation of our method was clearly recognized.
>
> Below we clarify the three points listed under the weaknesses section.
>
> ## **1. VAE decoding cost**
>
> **We respectfully disagree that the proposed method introduces an expensive additional VAE decoding step.**
>
> Our approach *does not* require any extra VAE decoding beyond what is already present in the baseline training pipeline. While ControlNet++ [1] and CTRL-U [2], upon which our baseline implementation is built, rely on VAE decoding, the alignment loss we introduce operates solely on UNet features via lightweight prediction heads. These heads are small (~8.5M parameters) and add only ~0.01 ms per prediction, resulting in negligible computational and memory overhead.
> Moreover, our method can be directly applied to the original ControlNet training pipeline, which does not involve VAE decoding at all. In this setting, our method still improves controllability while maintaining image quality (33.95 → 32.80 RMSE, 18.61 → 18.55 FID).
>
> ## **2. Contribution relative to ControlNet++**
>
> **We respectfully disagree that our contribution is limited to “step-level feedback” or represents a trivial extension of ControlNet++.**
>
> Our contribution is more fundamental and addresses a key gap in existing training pipelines:
>
> - **Early-stage control alignment:** we introduce a new training objective that enforces consistency between the input control signal (depth, edge, segmentation) and the structural information emerging in the **early diffusion stages**, where the global layout of the image is formed. Prior methods, including ControlNet++, focus exclusively on final-step feedback, overlooking these early layers where alignment is most critical.
>
> - **Enhanced controllability beyond reward-based training:** our alignment strategy strengthens control consistency across the full denoising trajectory and leads to improved controllability and image quality across diverse spatial tasks. This goes beyond ControlNet++ and CTRL-U, which rely on reward-based feedback limited to final outputs.
>
> Thus, our method is not a minor or trivial modification; it introduces a novel perspective on how structural consistency should be learned during diffusion denoising.
>
> ## **3. Magnitude of improvements**
>
> **We respectfully disagree that the improvement is marginal.**
>
> When built upon ControlNet++, our method yields consistent and measurable gains, especially for tasks dependent on global structure:
>
> - Depth: 27.63 → 26.09 RMSE, 18.59 → 18.29 FID
>
> - Segmentation: 38.08 → 40.22 CE, 39.04 → 37.65 FID
>
> The improvement on edge-based maps is smaller, which is expected since our alignment loss primarily acts on early denoising steps where global layout (depth, segmentation) is established. Fine edge information emerges later in the denoising process, making it inherently less responsive to early-stage alignment.
>
> Thus, the observed differences are consistent with the underlying generative dynamics rather than a limitation of our approach.
>
> Additionally, we would like to thank the reviewer for **his question**. We would like to clarify that ControlNet actually produces a better FID. We will make this correction in our main text.
>
> ## References
> [1] Ming Li, Taojiannan Yang, Huafeng Kuang, Jie Wu, Zhaoning Wang, Xuefeng Xiao, and Chen Chen. Controlnet++: Improving conditional controls with efficient consistency feedback: Project page: liming-ai. github. io/controlnet plus plus. In European Conference on Computer Vision, pp. 129–147. Springer, 2024
> [2] Guiyu Zhang, Huan-ang Gao, Zijian Jiang, Hao Zhao, and Zhedong Zheng. Ctrl-u: Robust conditional image generation via uncertainty-aware reward modeling. arXiv preprint arXiv:2410.11236, 2024
> [3] Yida Chen, Fernanda Vi´egas, and Martin Wattenberg. Beyond surface statistics: Scene representations in a latent diffusion model, 2023
> [4] Dmitry Baranchuk, Ivan Rubachev, Andrey Voynov, Valentin Khrulkov, and Artem Babenko. Label- efficient semantic segmentation with diffusion models. arXiv preprint arXiv:2112.03126, 2021

---

### Author Response · Authors · 2025-12-03
**Summary for Area Chairs (part 1/2)**

**Dear Area Chairs,**

We are submitting this summary to highlight the strengths of our work, recognized by the reviewers, and to present what experiments were conducted during the rebuttal addressing all provided questions.

## **Foundational strength**
We would like to emphasize a key strength identified by the reviewers:

- **Clear identification of prior limitations.** Our paper provides a precise explanation of why earlier reward-based losses (e.g., ControlNet++) fail, pinpointing the instability caused by one-step prediction signals.
Robust use of intermediate features. Leveraging UNet’s intermediate features through lightweight small networks provides a far more stable and accurate supervision signal in early denoising steps, enabling alignment across the full generation trajectory.

- **Complementary loss design.** The proposed alignment loss complements rather than replaces reward losses. This design is model-agnostic and integrates seamlessly with existing pipelines such as ControlNet and CTRL-U.

- **Improved control without quality degradation.** Across depth, edge, and segmentation tasks on MultiGen-20M and ADE20K, our method achieves better control metrics (RMSE, SSIM, mIoU) while maintaining image quality (FID).

- **Broad impact and modularity.** By enabling early-stage supervision and removing temporal limitations of prior reward losses, InnerControl provides a flexible, plug-and-play alignment loss with potential applications in controllable generation, video, and multi-modal editing.

- **Strong empirical validation.** Extensive experiments and ablations demonstrate consistent improvements and confirm the stability and accuracy of our probes across tasks, datasets, and training pipelines

## **Reviewers’ questions and our comprehensive experimental validation**
Throughout the rebuttal period, we conducted additional experiments demonstrating the broad applicability and effectiveness of our method. These experiments directly address central concerns raised across the reviews.

**Conclusive Validation of Generalizability**

- **Model-Agnostic Performance (addressing R3).** We validated that our method is not bound to a specific model. On the **SD-XL depth control task**, our alignment strategy improves control precision while preserving image quality: **RMSE:** 30.24 → 28.31; **FID:** 17.00 → 17.10.

- **Pipeline-Agnostic Performance.** Beyond the main paper’s results showing improvement ControlNet++ we prove that ControlNet also shows better results (33.95 RMSE, 18.61 FID → 32.80 RMSE, 18.55 FID). During rebattle we further demonstrate that **InnerControl also consistently improves the CTRL-U training pipeline:** RMSE: 26.50 → 25.40; FID: 18.67 → 18.56.

These results confirm that our method **generalizes** across **models** and **training protocols**.

---

> ### Author Response · Authors · 2025-12-03
> **Summary for Area Chairs (part 2/2)**
>
> ### **Other questions addressed**
>
> - **Minimal computational and memory overhead (addressing R1 & R3).** Our alignment networks are small (8.5M parameters vs. SD 1.5’s 980M) and introduce negligible overhead.
>
>  • Training speed: **0.46s → 0.47s** per iteration
>
>  • Memory usage: **+3GB**, well within typical hardware constraints.
>
> - **Simplified pre-training (addressing R2).** Firstly, we want to emphasize that our alignment nets contain only **8.5M** parameters and require around **2 hours** of training on a single V100 GPU. In our initial experiments, the models were trained separately. To simplify the pipeline, we **successfully trained a single multi-task alignment model** with a shared backbone (**9.49M** parameters) for HED, Depth, and LineArt, achieving performance identical to the individual probes. The corresponding results are included in the Appendix **Unified alignment model** (highlighted in blue).
>
> - **Loss formulation and timestep schedules (addressing R2 & R3).** We added a detailed explanation of our alignment timestep selection strategy. First, we compare alignment network vs. reward-model predictions across timesteps to estimate effective alignment regions. Then we perform a coarse grid search. The full procedure is included in the Appendix **Timesteps schedules**   (highlighted in blue).
>
> - **Reliability of intermediate features (addressing R3).** We directly addressed the question regarding the reliability of intermediate features compared to one-step prediction. While the main text already shows that our alignment models achieve better RMSE than reward models for depth estimation, we **additionally included the corresponding experiments for HED, LineArt, and Segmentation** in the Appendix **Figure 8**. These new results demonstrate that **the signal extracted from intermediate features is more stable and reliable during the initial generation steps across all tested control tasks.**
>
> - **Alternative samplers instead of one-step prediction (addressing R2).** We provide experiments in Appendix A2 **Alternative to one-step prediction** (highlighted in blue) showing that utilizing more complex samplers (such as DDIM) remains prohibitively expensive in both memory and time for models like SD 1.5.
>
> - **Using small networks during inference (addressing R3).**
> Our findings demonstrate that while the lightweight prediction networks are highly effective when integrated into the training-time alignment loss (as shown in our core results), applying them for on-the-fly inference correction reduces efficiency and does not improve output controllability. All corresponding results are included in the Appendix A2 **Utilizing small networks during inference** (highlighted in blue).
>
> ### **Regarding Reviewer 1**
> We also wish to note that Reviewer 1 raised no substantive technical questions. A key misunderstanding in the review was the assertion that InnerControl requires additional VAE decoding. This is **fundamentally incorrect.**
>
> - **Our alignment loss operates entirely on UNet features and requires no VAE decoding**. This fast forward step based on diffusion features is, in fact, one of the primary **strengths** of the proposed pipeline, and the lightweight nature of our small alignment models ensures a fast forward pass.
>
> - **We addressed all other mentioned weaknesses** (magnitude of improvement, scope of contribution) with clear evidence and explanations, demonstrating that the initial assessment did not fully reflect an accurate understanding of our work.
>
> **We hope this clarifies the robustness, generality, and technical soundness of our contribution, and we appreciate your consideration.**

---

### Meta-Review · Area_Chair_7drq · 2026-01-07

**Summary:**

This work combines existing Readout Guidance with ControlNet++ to address early-stage instability during original ControlNet++ training.
Given the following concerns, the AC is aligned with Reviewer 5Wbu (rating 4) and believes substantial efforts would be needed to improve the work. The AC recommends a weak rejection in this submission but with encouragement to resubmit to a future venue.

**Outstanding concerns**:

1. **Limited technical novelty**: The core method can be interpreted as a training-time variant of Readout Guidance, which was originally proposed for inference-time control. As a result, the contribution appears closer to an engineering combination of Readout Guidance and ControlNet++ rather than a fundamentally new alignment framework.
2. **Control-specific training requirement**: The approach inherits a key limitation noted by multiple reviewers, namely the need to train separate probe models for each control type. This restricts scalability and limits applicability to mixed or heterogeneous control inputs.
3. **Limited scope of evaluation**: All experiments focus on dense, pixel-wise controls. The lack of experiments on sparse controls (e.g., keypoints or bounding boxes), as raised by Reviewer 5Wbu, leaves open questions about the generality of the approach.

**Reviewer Concerns:**

**Concerns addressed by the rebuttal:**
- **Early-step instability in ControlNet++:** The rebuttal and additional experiments clearly support the claim that intermediate diffusion features provide a more stable supervision signal than one-step predictions, effectively resolving early-stage instability.


**Concerns that remain outstanding:**
- **Per-control training requirement:** The need to train control-specific probes remains and limits scalability and applicability to mixed-control settings.
- **Limited evaluation scope:** The absence of experiments on sparse control signals (e.g., keypoints, bounding boxes) remains an open gap, as noted by Reviewer 5Wbu.

**Reviewer Scores:**

| Reviewer | Initial Score | AC Estimated Score | AC Reason |
|--------|---------------|-------------------|-----------|
| PrLh | - | - | **This reviewer was ignored by AC due to very low-quality reviews**|
| e6tR | 6 | 6 | - |
| 5Wbu | 4 | 4 | Similarity to Readout Guidance; Limited scope of evaluation|

---

### Decision · Program_Chairs · 2026-01-26

Reject